# The hydrophobic nature of a novel membrane interface regulates the enzyme activity of a voltage-sensing phosphatase

**Akira Kawanabe[1], Masaki Hashimoto[2], Manami Nishizawa[3], Kazuhisa Nishizawa[3], Hirotaka Narita[4], Tomoko Yonezawa[1], Yuka Jinno[1], Souhei Sakata[5], Atsushi Nakagawa[4], Yasushi Okamura[1,2]\***

[1]Integrative Physiology, Department of Physiology, Graduate School of Medicine, Osaka University, Osaka, Japan; [2]Graduate School of Frontier Biosciences, Osaka University, Osaka, Japan; [3]School of Medical Technology, Teikyo University, Tokyo, Japan; [4]Institute for Protein Research, Osaka University, Osaka, Japan; [5]Department of Physiology, Division of Life Sciences, Faculty of Medicine, Osaka Medical College, Osaka, Japan

**Abstract** Voltage-sensing phosphatases (VSP) contain a voltage sensor domain (VSD) similar to that of voltage-gated ion channels but lack a pore-gate domain. A VSD in a VSP regulates the cytoplasmic catalytic region (CCR). However, the mechanisms by which the VSD couples to the CCR remain elusive. Here we report a membrane interface (named 'the hydrophobic spine'), which is essential for the coupling of the VSD and CCR. Our molecular dynamics simulations suggest that the hydrophobic spine of *Ciona intestinalis* VSP (Ci-VSP) provides a hinge-like motion for the CCR through the loose membrane association of the phosphatase domain. Electrophysiological experiments indicate that the voltage-dependent phosphatase activity of Ci-VSP depends on the hydrophobicity and presence of an aromatic ring in the hydrophobic spine. Analysis of conformational changes in the VSD and CCR suggests that the VSP has two states with distinct enzyme activities and that the second transition depends on the hydrophobic spine.
DOI: https://doi.org/10.7554/eLife.41653.001

**\*For correspondence:**
yokamura@phys2.med.osaka-u.ac.jp

**Competing interests:** The authors declare that no competing interests exist.

## Introduction

Changes in membrane potential induce structural changes in membrane proteins such as ion channels, transporters, and receptors, and underlie membrane excitability, neurotransmitter and hormone release, muscle contraction (*Proenza et al., 2002*), cell proliferation (*Jehle et al., 2011*), and reactive oxygen species production (*DeCoursey, 2013*). Voltage-sensor domains (VSDs) are sophisticated protein modules that sense changes in transmembrane voltage and regulate downstream effectors. VSDs consist of four transmembrane helices and a fourth helix that contains a signature pattern of positively charged residues that align to form salt bridges with acidic residues on the other helices. In classical voltage-gated ion channels (VGICs), VSDs are connected to a pore-gate domain (PGD) composed of two transmembrane helices that form an ion-permeation pathway.

Voltage sensing phosphatases (VSPs) belong to a family of enzymes that include phosphoinositide phosphatase and tensin homolog deleted on chromosome ten (PTEN). VSPs contain a VSD and PTEN-like cytoplasmic catalytic region (CCR), which consists of a phosphatase domain (PD) and C2 domain (*Murata et al., 2005*; *Okamura et al., 2018*). The motion of the VSD in response to membrane depolarization activates the enzymatic activity of the CCR, leading to dephosphorylation of

phosphoinositides, mainly PI(4,5)P$_2$, and providing a direct pathway to translate electric signals to chemical signals (*Murata and Okamura, 2007*; *Sakata et al., 2011*; *Sakata and Okamura, 2014*; *Okamura et al., 2018*). The structures of both the VSD and CCR of *Ciona intestinalis*-VSP (Ci-VSP) were individually solved by X-ray crystallography (*Matsuda et al., 2011*; *Liu et al., 2012*; *Li et al., 2014*). Compared to VGICs, VSPs show two contrasting features. First, the motion of the VSD activates the cytoplasmic structure that dephosphorylates phosphoinositides, unlike the action of VGICs, which ultimately results in structural changes in the transmembrane PGD. Second, a single VSD regulates a single CCR, in contrast with VGICs, in which four homologous units (repeat or subunit) assemble to regulate a central pore gate in a complex manner. Although a recent study suggested that VSPs can form dimers with high density expression and that dimeric VSPs exhibit slightly different molecular properties than those of monomers, the basic nature of voltage-dependent enzyme activity is innate to monomers (*Rayaprolu et al., 2018*).

One of the central questions about VSPs is how the VSD motion triggers the catalytic activity of the CCR. Two major findings provide clues to the mechanisms underlying VSD and CCR coupling. First, deletion or mutation of the linker between VSD and the phosphatase domain (which is called 'the VSD-PD linker') eliminates coupling (*Murata et al., 2005*; *Villalba-Galea et al., 2009*; *Kohout et al., 2010*). The crystal structure of the CCR of Ci-VSP (*Liu et al., 2012*) showed that two residues of Lys-252 and Arg-253 directly interact with residues in the enzyme active center. Based on multiple crystal structures of the CCR, Liu et al. proposed a model where a loop region, called a 'gating loop,' which is not present in PTEN, moves dynamically upon VSD activation to regulate steric hindrance in substrate binding. These reports suggest that the VSD-PD linker of the VSP works as a hub to translate information from the movement of VSD to a conformational change in the CCR.

Another important clue about the mechanisms of VSD-CCR coupling is the multiple active states of the enzyme, which are coupled with distinct states of the VSD. A zebrafish VSP with a double mutation in the S4 domain, Dr-VSP (T156R/I165R), exhibits two-step activation of the VSD, and its phosphatase activity depends on the extent of VSD motion (*Sakata and Okamura, 2014*), suggesting that enzyme activity can be tuned by the extent of motion of the VSD at the single protein level. In another study, the CCR phosphatase activity of several VSD mutants was analyzed by fast-responding phosphoinositide (PIP) sensors (*Grimm and Isacoff, 2016*), and the results showed that the distinct activation states of the VSD induce the activation of enzymes with distinct substrate preferences. Hille's group showed, in a simple two-state transition model for activation of the enzyme, a reconstituted time course of voltage-evoked signaling of a fast fluorescent reporter sensitive to PIPs (*Keum et al., 2016*). Our recent analysis of a version of Ci-VSP containing an environmentally sensitive unnatural fluorescent amino acid, Anap, at Lys-555 showed a biphasic change in Anap fluorescence that was dependent on the membrane voltage: the Ci-VSP exhibited a decrease in fluorescence at low membrane voltage and then an increase at higher membrane voltages, consistent with the notion that the CCR in Ci-VSP has multiple conformations (*Sakata et al., 2016*). However, the mechanisms underlying the adoption of these multiple conformations of CCR upon VSD activation remain unclear.

In this study, we attempted to gain more mechanistic insights into voltage-induced activation of the CCR. To achieve this goal, we combined coarse-grained (CG) and atomistic (AT) molecular dynamics (MD) simulation analyses of the interaction between the whole CCR and the membrane with biophysical characterization of the conformational changes of the VSD and CCR in *Xenopus* oocytes. MD simulations revealed a novel site in the PD with two hydrophobic residues (here, named 'the hydrophobic spine'; Leu-284 and Phe-285 in Ci-VSP) that protrude from the CCR and contact the membrane. AT MD simulations suggested a 'compromised nature' of the hydrophobic spine; despite hydrophobicity that should favor penetration into the hydrophobic core of the membrane (*MacCallum et al., 2008*), the simulations showed its anomalously loose association with the bilayer surface, shallow positioning in the bilayer, and high mobility in and near the phospholipid head group layer. Electrophysiological studies of numerous versions of Ci-VSP with mutations in the two residues showed that hydrophobicity of the residues is critical for voltage-dependent phosphatase activity, and that an aromatic residue on the hydrophobic spine facilitates VSD and CCR coupling. A voltage-clamp fluorometry analysis with cysteine-based incorporation of a fluorophore on the VSD and genetic incorporation of Anap into the CCR indicated that the CCR is activated by the VSD via two steps and that the transition from an intermediate state to a fully-activated state involves the hydrophobic spine.

## Results

### MD simulation of the cytoplasmic catalytic region of VSP reveals the possible function of the VSP hydrophobic spine

The voltage-sensing phosphatase (VSP) consists of a voltage sensor domain (VSD) and a cytoplasmic catalytic region (CCR). The latter is composed of three major regions: the linker (the VSD-PD linker), the phosphatase domain (PD), and the C2 domain (*Figure 1*).

To examine the overall propensity of the CCR of the VSP to bind to phospholipid bilayers, we first performed twelve 2-μs CG MD simulations in a system made of a 1-palmitoyl-2-oleoyl-glycero-3-phosphocholine (POPC) bilayer, a water layer, and the CCR of *Ciona intestinalis*-VSP (Ci-VSP), whose crystal structure has already been determined (*Matsuda et al., 2011*; *Liu et al., 2012*). The CCR was initially placed in four distinct orientations/positions (cg_po_1–4 in *Table 1*, *Figure 2—figure supplement 1*), with its center of mass placed ~3.5–4.0 nm away from the plane and containing the phosphorus atoms of the proximal monolayer ('phosphorus layer'). Neither the C2 domain nor the PD formed a stable complex with the bilayer (*Figure 2—figure supplement 2A–D* and *Table 1*). To determine the potential for the CCR to weakly bind to the bilayer, we performed 1-μs CG runs starting with the CCR pre-bound to the POPC bilayer (cg_po_5a to 5 hr) in the productive orientation ('The productive orientation' and 'the protein binding to the bilayers' are defined in the section of MD simulation in Materials and methods and *Figure 2—figure supplement 3*). In all eight runs, the PD detached from the bilayer, but, intriguingly, in four runs, the C2 domain remained bound throughout the simulations (*Figure 2—figure supplement 4*). This result raised the possibility that the C2 domain has an affinity for the POPC bilayer that is stronger than that for the PD. When

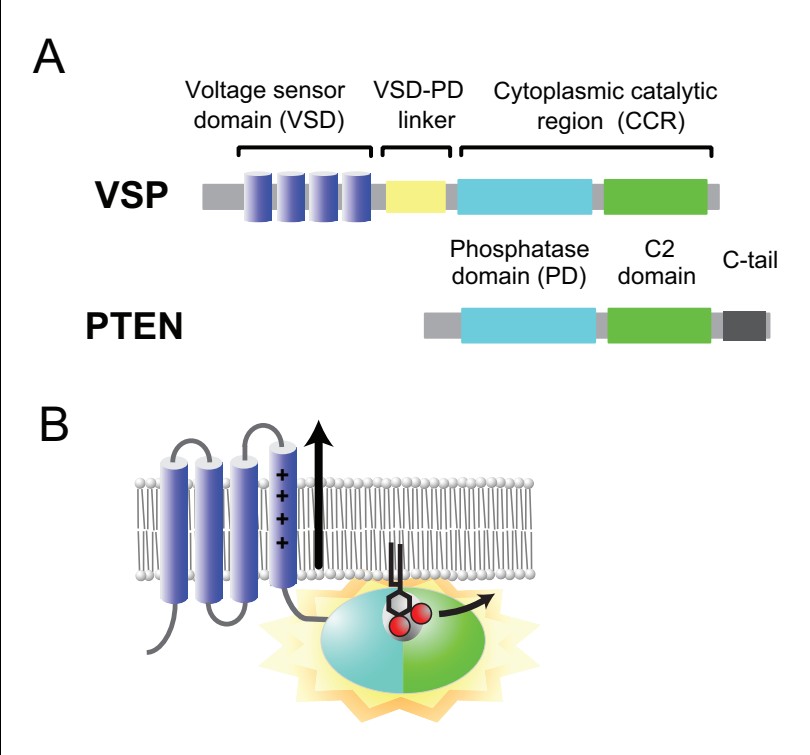

**Figure 1.** Voltage-sensing phosphatase. (**A**) Domain architecture of VSP and PTEN. The VSP consists of a voltage sensor domain (VSD, dark blue), VSD-PD linker (light yellow), and a cytoplasmic catalytic region [CCR, cyan (phosphatase domain) and green (C2 domain)]. The CCR of the VSP shows high sequence similarity to that of PTEN, except that a C-terminal tail is absent (C-tail, dark gray). (**B**) The VSD is activated by membrane depolarization, and the motion induces conformational changes in the CCR through the VSD-PD linker. The black sticks with two red spheres show a substrate, PI(4,5)P$_2$.
DOI: https://doi.org/10.7554/eLife.41653.002

**Table 1.** Summary table of simulations.

| Simulation ID | Bilayer composition | Duration | Initial position (nm) / orientation[*] | Time fraction (%) showing PD-bilayer binding[†] | Time fraction (%) showing C2-bilayer binding[†] | Time fraction (%) showing the productive orientation[†,‡] |
|---|---|---|---|---|---|---|
| *coarse-grained* | | | | | | |
| cg_po_1a, b, c | 212 POPC | 3 × 2μs | −3.5 / C2$^{K558}$ | 0.0, 0.0, 0.8 | 10.8, 0.0, 0.0 | 0.0, 0.0, 0.0 |
| cg_po_2a, b, c | 212 POPC | 3 × 2μs | −4.1 / LD$^{R246}$ | 0.4, 0.0, 0.0 | 0.0, 0.0, 0.0 | 0.0, 0.0, 0.0 |
| cg_po_3a, b, c | 212 POPC | 3 × 2μs | −3.5 / PD$^{K364}$, C2$^{Y522}$ | 0.0, 0.0, 0.0 | 0.0, 0.0, 0.8 | 0.0, 0.0, 0.0 |
| cg_po_4a, b, c | 212 POPC | 3 × 2μs | −3.5 / PD$^{K427}$, PD$^{Y429}$ | 0.0, 0.0, 0.0 | 0.0, 0.2, 0.0 | 0.0, 0.0, 0.0 |
| cg_po_5a to h | 212 POPC | 8 × 1μs | −1.6 / LD$^{R246}$, C2$^{K558}$ (prebound) | 4.4, 8.4 [§] | 45.8, 99.6 [§] | 2.8, 3.6 [§] |
| cg_po_pi_1a, b, c | 208 POPC/4 PI (3,4,5)P$_3$ | 3 × 2μs | −4.0 / C2$^{K558}$ | 95.8, 86.6, 93,4 | 95.2, 12.8, 44.7 | 71.7, 0.4, 34.7 |
| cg_po_pi_2a, b, c | 208 POPC/4 PI (3,4,5)P$_3$ | 3 × 2μs | −4.0 / LD$^{R246}$ | 99.6, 93.6, 97.6 | 5.0, 57.1, 38.7 | 3.8, 33.3, 3.2 |
| cg_po_pi_3a, b, c | 208 POPC/4 PI (3,4,5)P$_3$ | 3 × 2μs | −4.0 / PD$^{K364}$, C2$^{Y522}$ | 78.8, 86.6, 72.9 | 50.7, 48.7, 98.8 | 30.7, 27.3, 34.7 |
| cg_po_pi_4a, b, c | 208 POPC/4 PI (3,4,5)P$_3$ | 3 × 2μs | −3.5 / PD$^{K427}$, PD$^{Y429}$ | 83.2, 89.8, 90.6 | 13.0, 4.6, 28.9 | 0.0, 3.4, 7.4 |
| *Atomistic* | | | | | | |
| at_po_1 | 212 POPC | 100ns | −2.6 / LD$^{R246}$, C2$^{K558}$ | 42.2 | 97.1 | 42.2 |
| at_po_2 | 212 POPC | 100ns | −2.6 / LD$^{R246}$, C2$^{K558}$ | 92.5 | 99.0 | 51.0 |
| at_po_3 | 212 POPC | 100ns | −2.6 / LD$^{R246}$, C2$^{K558}$ | 97.2 | 98.2 | 97.2 |
| at_po_4 | 212 POPC | 100ns | −3.2 / LD$^{R246}$, C2$^{K558}$ | 98.5 | 99.5 | 97.5 |
| at_po_5 | 212 POPC | 100ns | −3.2 / LD$^{R246}$, C2$^{K558}$ | 10.4 | 85.1 | 0.0 |
| at_po_6 | 212 POPC | 100ns | −3.2 / LD$^{R246}$, C2$^{K558}$ | 0.5 | 90.7 | 0.0 |
| at_po_pi_0 [¶] | 208 POPC/4 PI (3,4,5)P$_3$ | 100ns | −2.5 / PD$^{R281}$, C2$^{K558}$ | 97.9 | 39.1 | 39.1 |
| at_po_pi_1 | 208 POPC/4 PI (3,4,5)P$_3$ | 100ns | −3.1 / LD$^{R246}$, C2$^{K558}$ | 86.1 | 81.2 | 60.9 |
| at_po_pi_2 | 208 POPC/4 PI (3,4,5)P$_3$ | 100ns | −3.1 / LD$^{R246}$, C2$^{K558}$ | 91.1 | 31.2 | 28.2 |
| at_po_pi_3 | 208 POPC/4 PI (3,4,5)P$_3$ | 100ns | −3.4 / LD$^{R246}$, C2$^{K558}$ | 79.7 | 89.1 | 77.2 |
| at_po_pi_4 | 208 POPC/4 PI (3,4,5)P$_3$ | 100ns | −3.4/ LD$^{R246}$, C2$^{K558}$ | 0.0 | 80.0 | 0.0 |
| at_po_pi_5 | 208 POPC/4 PI (3,4,5)P$_3$ | 100ns | −3.1 / PD$^{R281}$, C2$^{K516}$ | 94.5 | 19.9 | 17.9 |
| at_po_pi_6 | 208 POPC/4 PI (3,4,5)P$_3$ | 100ns | −3.4 / PD$^{R281}$, C2$^{K516}$ | 84.1 | 95.5 | 83.6 |
| at_po_pi_7 | 208 POPC/4 PI (3,4,5)P$_3$ | 100ns | −3.4 / PD$^{R281}$, C2$^{K516}$ | 85.1 | 98.1 | 84.6 |
| at_steered | 208 POPC/4 PI (3,4,5)P$_3$/ 212 POPC, | Two runs named POPC/PI(3,4,5)P$_3$_1 and 2 were started with structures sampled from at_po_pi_5 and POPC_1 and 2) were with two structures from at_po_4.[#] | | | | |

[*]The protein position is represented by the z-position (i.e., the positions along the z-axis, which is parallel to the membrane normal) of the center of mass of the protein relative to the phosphorus layer. Orientation is represented by the domain name [C2, PD or VSD-PD linker (LD)] along with, in superscript,

the residue initially located the nearest to the bilayer. 'PD$^{K364}$, C2$^{Y522}$' for example, represents that both Tyr-522 of the C2 domain and Lys-364 of the PD were equally close to the bilayer.

†For the CG sets, three values correspond to the three independent runs (a,b and c).

‡The fraction of the time period showing not only the productive orientation but also the binding of the both the PD and C2 domain is shown.

§The results of two representative runs are shown.

¶Initial structure was prepared mimicking the structure sampled from cg_po_pi_1a. In this run, both domains were initially set bound to the POPC/PI(3,4,5)P$_3$ bilayer, but the C2 domain detached from and the PD remained bound to the bilayer.

#The legend for **Figure 2—figure supplement 6** has more detail.

DOI: https://doi.org/10.7554/eLife.41653.003

POPC/PI(3,4,5)P$_3$ bilayers, instead of the POPC bilayer, were used, most runs showed the formation of CCR-bilayer complexes (**Figure 2—figure supplement 2E–H**), in agreement with a report on the stabilization of the CCR-bilayer complex with the addition of PI(3,4,5)P$_3$ to the POPC bilayer (**Kalli et al., 2014**). The effect of PI(3,4,5)P$_3$ appeared more pronounced for PD binding than for binding of the C2 domain to the bilayer (**Figure 2—figure supplement 2G,H**).

We next performed atomistic (AT) MD simulations. For the POPC sets (at_po_1 to 6 of **Table 1**), the center of mass of CCR was initially placed 2.6 or 3.2 nm away from the phosphorus layer of the POPC bilayer in the productive orientation. The time of the z-positions (i.e., positions along the z-axis, which is parallel to the membrane normal) of the C2 domain and PD showed membrane binding that was different between the two domains (at_po_1 to 6 in **Table 1**). The C2 domain quickly bound to the POPC bilayer in all runs, but the PD slowly approached the bilayer over 50–80 ns in three of the six runs, whereas it approached the bilayer relatively quickly in the remaining three runs (**Figure 2—figure supplement 5A–C**). **Figure 2** demonstrates two representative runs; one (at_po_4) that showed binding of both domains (blue and black lines indicating the C2 domain and PD, respectively) and another (at_po_5) that showed binding of the C2 domain but not of PD (light blue and grey lines). When four PI(3,4,5)P$_3$ molecules were randomly added in the lipid monolayer facing the Ci-VSP of the POPC bilayer, binding of both domains to the bilayer were expedited (at_po_pi_1 to 7) (**Figure 2—figure supplement 5D–F** and **Table 1**). Overall, the presence of PI(3,4,5)P$_3$ appeared to enhance the affinity of the CCR to the membrane; however, this enhancement effect was more evident for PD. Thus, our simulation data suggested that the C2 domain had substantial affinity for the POPC bilayers in the absence of PI(3,4,5)P$_3$ molecules and that the presence of PI(3,4,5)P$_3$ can increase affinity of the PD to phospholipid bilayers.

Given the plausible effect of the VSD linkage to limit wide excursions of the CCR from membranes, the above results suggest that the C2 domain constitutively binds the bilayer and that the distance of the PD from the bilayer can be altered through rotational and/or hinge-like motions of the CCR in response to both structural changes in the VSD and the concentration of PI(3,4,5)P$_3$ molecules. The loop containing Leu-284 and Phe-285, which we call the 'hydrophobic spine,' was located near the distal end of this putative rotational motion and served as the point of contact between the PD and the bilayer (**Figures 2** and **3B**). In addition to its membrane-facing location, the hydrophobic spine attracted our attention because it appeared compromised. Specifically, despite its hydrophobic property, which should have favored penetration into the hydrophobic core of the phospholipid bilayer, neighboring residues were polar or ionizable (**Figure 3A**), likely hampering this penetration. In support of this, our steered MD simulations, in which the hydrophobic spine was pulled upward with an external force and forced into the phosphorus layer, resulted in the hydrophobic spine bouncing back to shallower positions when the pulling force was turned off (**Figure 2—figure supplement 6**), suggesting that the spine cannot penetrate the hydrophobic core of the bilayer, likely because of the hydrophilicity of residues near the spine. During the period when the PD apparently associated with membrane, the spine residues (Leu-284 and Phe-285) showed a superficial positioning that suggested a loose association with the bilayer, with their center of mass located ~3–4 Å from the phosphorus layer, and the z-position showing fluctuations in the range of ~3 Å (**Figure 2—figure supplement 6**). Overall, the AT simulations suggested that association of the PD with the POPC bilayer was fairly weak, allowing for fluctuations along the z-position and occasional detachment of the PD from the bilayer.

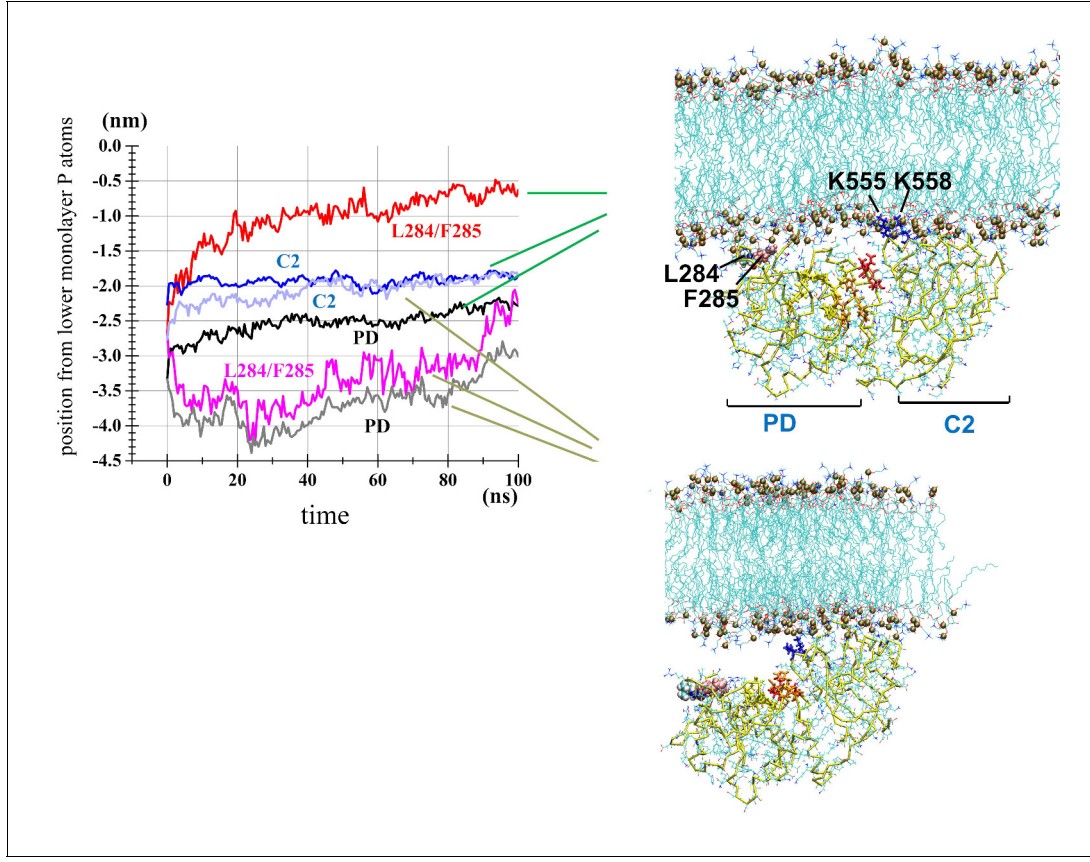

**Figure 2.** MD simulation of the cytoplasmic catalytic region. Two selected AT runs. Left: Time courses of the z-positions of the centers of mass of the C2 domain, PD, and hydrophobic spine (Leu-284 and Phe-285) for the run at_po_4, which showed rapid membrane binding of both domains (blue, black, and red lines show the C2 domain, PD, and hydrophobic spine, respectively) and at_po_5, which exhibited rapid C2 domain and slow PD binding (light blue, grey, and pink lines show the C2 domain, PD, and hydrophobic spine, respectively). Of note, 'po' of the simulation IDs stands for the POPC bilayer. Right top: a snapshot of the time frame at 80 ns on the trajectory of at_po_4. Leu-284 and Phe-285 are shown with van der Waals forces represented and Phe-285 shown in pink. The following residues of the catalytic center are colored: His-330 to Asn-333 are shown in orange, and Val-410 to Thr-412 are shown in red. Right bottom: snapshot at 80 ns on the trajectory of at_po_5.

DOI: https://doi.org/10.7554/eLife.41653.004

The following figure supplements are available for figure 2:

**Figure supplement 1.** Orientation of the structure of the cytoplasmic catalytic region used for simulations
DOI: https://doi.org/10.7554/eLife.41653.005

**Figure supplement 2.** Progress of the CG simulations started with the CCR placed away from the bilayers.
DOI: https://doi.org/10.7554/eLife.41653.006

**Figure supplement 3.** The productive orientation of the CCR.
DOI: https://doi.org/10.7554/eLife.41653.007

**Figure supplement 4.** Progress of CG simulation runs in the prebound state.
DOI: https://doi.org/10.7554/eLife.41653.008

**Figure supplement 5.** Progress of AT simulation runs (1).
DOI: https://doi.org/10.7554/eLife.41653.009

**Figure supplement 6.** Progress of AT simulation runs (2): steered simulations.
DOI: https://doi.org/10.7554/eLife.41653.010

## Mutations in 'the hydrophobic spine' affect voltage-dependent enzymatic activity

The functional role of the membrane-interacting region, 'the hydrophobic spine,' was addressed by estimating the voltage-dependent enzymatic activities of Ci-VSP mutants heterologously expressed in *Xenopus* oocytes. An inward rectifier K$^+$ channel (mouse-Kir3.2) was used for the read out of the

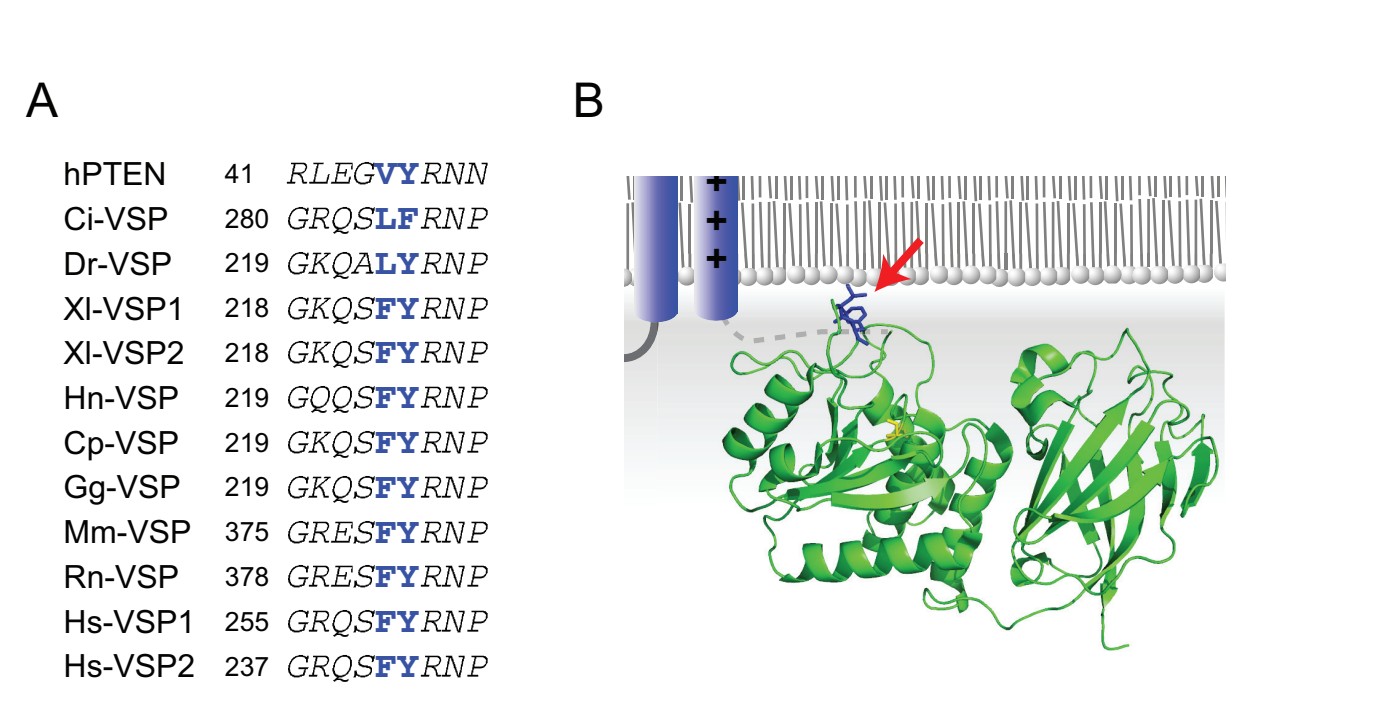

**Figure 3.** The hydrophobic spine. (**A**) Amino acid sequences of PTEN (human) and VSP orthologs around the hydrophobic spine. Ci: sea squirt (*Ciona intestinalis*), Cp: newt (*Cynops pyrrhogaster*), Dr: zebrafish (*Danio rerio*), Gg: chicken (*Gallus gallus*), Hn: salamander (*Hynobius nebulosus*), Hs: human (*Homo sapiens*), Mm: mouse (*Mus musculus*), Rn: rat (*Rattus norvegicus*), Xl: clawed frog (*Xenopus laevis*). (**B**) Putative conformation of voltage-sensing phosphatase near a phospholipid membrane (dark blue: voltage sensor domain; green: cytoplasmic catalytic region [X-ray crystal structure: 3V0H]). Blue amino acids indicated by red arrows are hydrophobic residues at the phosphatase domain surface (defined as the 'hydrophobic spine').
DOI: https://doi.org/10.7554/eLife.41653.011

voltage-dependent phosphatase activity toward PI(4,5)P$_2$ on the plasma membrane, as previously established (*Murata et al., 2005*).

The top panel of *Figure 4A* shows representative current traces, recorded by two-electrode voltage clamp, of PI(4,5)P$_2$-dependent Kir co-expressed with wild-type (WT) Ci-VSP. The pulse protocol is composed of three parts (*Figure 4A*, bottom panel): (1) a 50 ms ramp pulse to confirm the inward current from the Kir channel, (2) a 50 ms step pulse at −120 mV to detect steady-state Kir current amplitude (test pulse), and (3) a 300 ms depolarizing step pulse to 50 mV to activate Ci-VSP. This protocol was repeated 20 times. A large Kir current was detected in the first episode during a test pulse at −120 mV, followed by a clear decrease in the Kir current amplitude with repeated stimulation. The normalized Kir currents at the end of the 50 ms test pulse were plotted against the accumulated depolarization time (e.g., 1 st: 0 ms; 2nd: 300 ms) (*Figure 4B*, top panel). The decay kinetics of the time-dependent normalized Kir current in this plot indicates voltage-dependent phosphatase activity.

Leu-284 was mutated to Gly, Ala, Val, Thr, His, Phe, Tyr, Trp, Gln, Asp, and Lys. All mutant data obtained using the Kir method are shown in *Figure 4* and *Figure 4—figure supplements 1* and *2*. A time course of the normalized current amplitude of L284V overlapped with that of WT Ci-VSP, indicating that L284V results in voltage-dependent phosphatase activity similar to that of the WT. A phenylalanine substitution at Leu-284 (L284F) showed rapid Kir current decay, suggesting enhancement of the voltage-dependent phosphatase activity. A hydrophilic amino acid mutant, L284Q, showed relatively slow kinetics of Kir current decay, indicating that L284Q has low voltage-dependent phosphatase activity. The voltage-dependent Kir current decays of Ci-VSP hydrophobic spine mutants were fitted by a single exponential function and rate constants were obtained (*Figure 5A*).

We also mutated Phe-285 into Gly, Ala, Leu, Thr, His, Tyr, Trp, Asn, Asp, and Lys (*Figure 4*, *Figure 4—figure supplements 1* and *2*). A mutation to hydrophilic amino acid (F285Q) attenuates

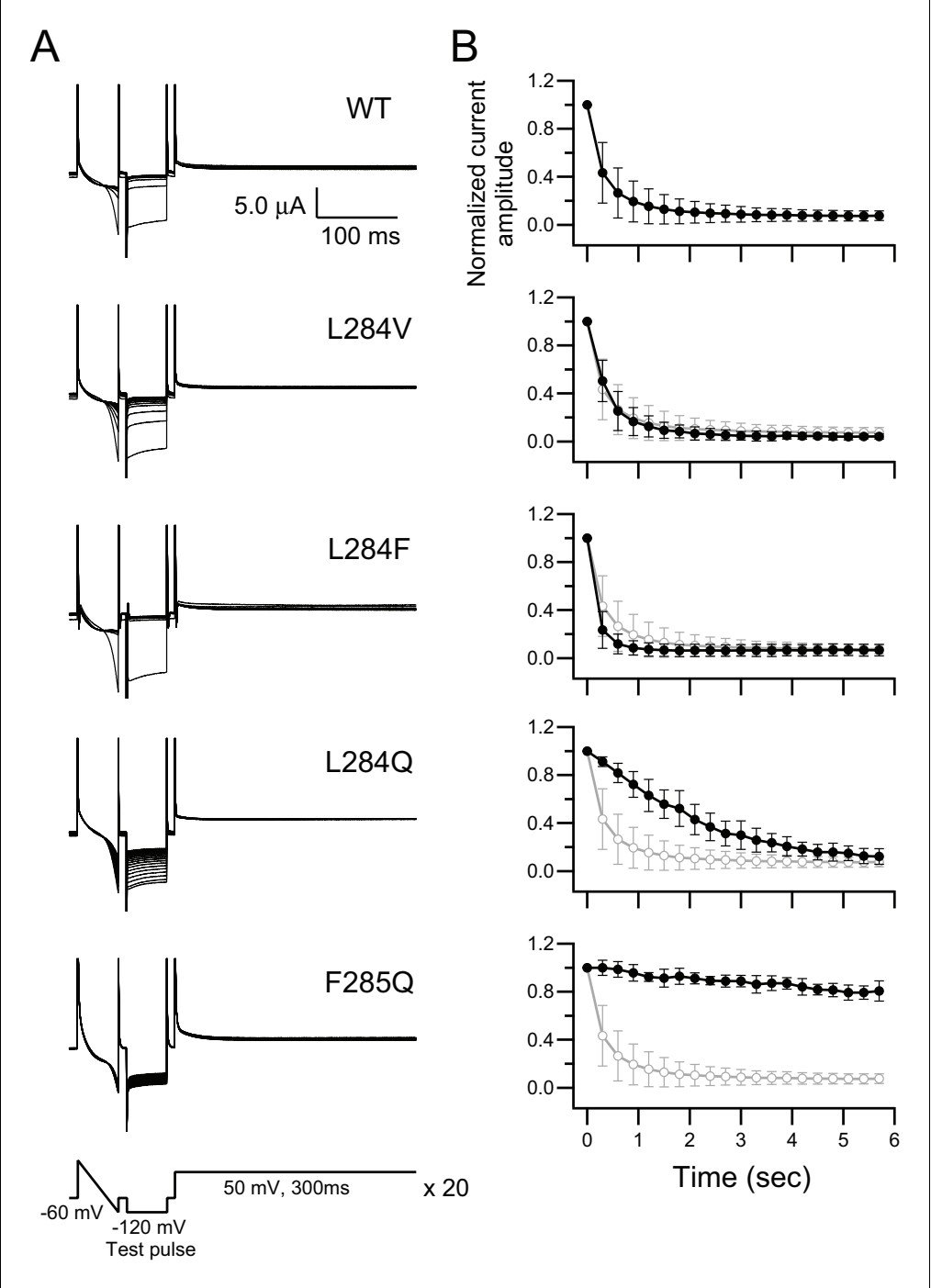

**Figure 4.** Alteration of voltage-dependent phosphatase activities by mutations in the hydrophobic spine. (**A**) Representative current traces of a PI(4,5)P$_2$-sensitive K$^+$ channel (Kir3.2d, GIRK2) coexpressed with Ci-VSP WT, L284V, L284F, L284Q, and F285Q in *Xenopus* oocytes. A pulse protocol (bottom panel) consists of a 50 ms ramp pulse, 50 ms step pulse at −120 mV (test pulse), and 300 ms depolarization pulse at 50 mV. This protocol was repeated 20 times. All 20 current traces are superimposed. The holding potential was maintained at −60 mV. Kir currents were observed in test pulses, and the current amplitude indicated the relative amount of PI(4,5)P$_2$ in the plasma membrane. (**B**) Plots of time-dependent normalized current amplitudes of inward K$^+$ currents at the end points of the test pulses. Values were normalized to the first trace current amplitude of the test pulse. Black lines show the plots from current traces of A. Gray lines show the plot from the WT, which is the same trace as that

*Figure 4 continued on next page*

*Figure 4 continued*

shown at the top. Symbols show the mean ± S.D. of data from 4 to 56 oocytes (WT: n = 56, L284V: n = 5, L284F: n = 35, L284Q: n = 4, F285Q: n = 4).

DOI: https://doi.org/10.7554/eLife.41653.012

The following source data and figure supplements are available for figure 4:

**Source data 1.** Numerical data that were used to generate the chart in *Figure 4B*.

DOI: https://doi.org/10.7554/eLife.41653.015

**Figure supplement 1.** A gallery of voltage dependent phosphatase activities of all Ci-VSP mutants examined with Kir channel (Representative current traces).

DOI: https://doi.org/10.7554/eLife.41653.013

**Figure supplement 2.** Pooled data of voltage dependent phosphatase activities of all Ci-VSP mutants examined with Kir channel (Current decays).

DOI: https://doi.org/10.7554/eLife.41653.014

voltage–dependent phosphatase activity, as does a mutation at Leu-284 (*Figure 4*). With F285D and F285K mutations, voltage-dependent decay of the Kir currents could not be observed with a 50 mV depolarization pulse (*Figure 4—figure supplements 1* and *2*), but the Kir currents decayed with higher membrane voltages (e.g., 100 mV), indicating that voltage-dependent enzyme activities are drastically reduced but not completely eliminated by these mutations (data not shown).

To confirm the expression of Ci-VSP mutants in the plasma membrane, we recorded the 'sensing current' from the same oocytes in which the decay rate of Kir upon VSP activities was examined. The maximum moving charge of the off-sensing current (off-sensing charge) was estimated by applying a repolarization step from 150 mV to −60 mV, which reflects the expression level of Ci-VSP mutants on cell membranes. Sensing currents were observed in all constructs that were tested (*Figure 5—figure supplement 1*). In some mutants (e.g., L284A and L284D), relatively low cell surface expression was observed, suggesting that their voltage-dependent phosphatase activities could be underestimated.

The above results suggest that the voltage-dependent phosphatase activities of Ci-VSP mutants in the hydrophobic spine are closely associated with the hydrophobicity of amino acid side chains with Leu-284 and Phe-285. To examine this idea in more detail, we plotted the voltage-dependent phosphatase activities of individual mutants against the hydrophobicity of an amino acid side chain, which was introduced either at Leu-284 or Phe-285 (*Figure 5B and C*). The hydrophobicity values for the amino acid side chain were obtained from Hessa et al. (*Hessa et al., 2005*), which estimated a free energy of water-phase membrane transition of each amino acid embedded in an α-helical protein as hydrophobicity in a biological condition. *Figure 5B and C* show that Ci-VSP with a hydrophobic amino acid at position 284 or 285 has relatively high voltage-dependent phosphatase activity, whereas that of Ci-VSP with a polar or charged amino acid has low voltage-dependent phosphatase activity, indicating that the activity of Ci-VSP depends on the hydrophobicity of the side chain in the hydrophobic spine. On the other hand, the relationship between the voltage-dependent phosphatase activity and the size of the side chain (*Zamyatnin, 1972*) was not clear (*Figure 5—figure supplement 2*). Moreover, we found that mutation into a residue containing an aromatic ring led to higher voltage-dependent phosphatase activity compared to that with other hydrophobic amino acid residues.

## Addition of an aromatic ring to the hydrophobic spine enhances voltage-dependent phosphatase activity

Ci-VSP L284F exhibited sharper Kir current decay than WT (*Figure 4*), suggesting that the presence of an aromatic ring enhances voltage-dependent phosphatase activity. To confirm whether an aromatic ring is essential for this enhancement effect, we measured enhancement with other aromatic amino acid mutations (L284Y, L284W, F285Y, and F285W, *Figure 4—figure supplements 1* and *2*). All of these showed higher voltage-dependent phosphatase activity than that of WT (*Figure 5A*). It is difficult to estimate the Kir current decay kinetics in oocytes with robust enzymatic activity using some Ci-VSP mutants, because the Kir current decays too rapidly under a pulse protocol of 300 ms depolarization. As shown in *Figure 5*, when the Kir current amplitude in the second trace was less than 1% of that in the first trace, we rounded the decay time constant to 0.065 s (a rate constant

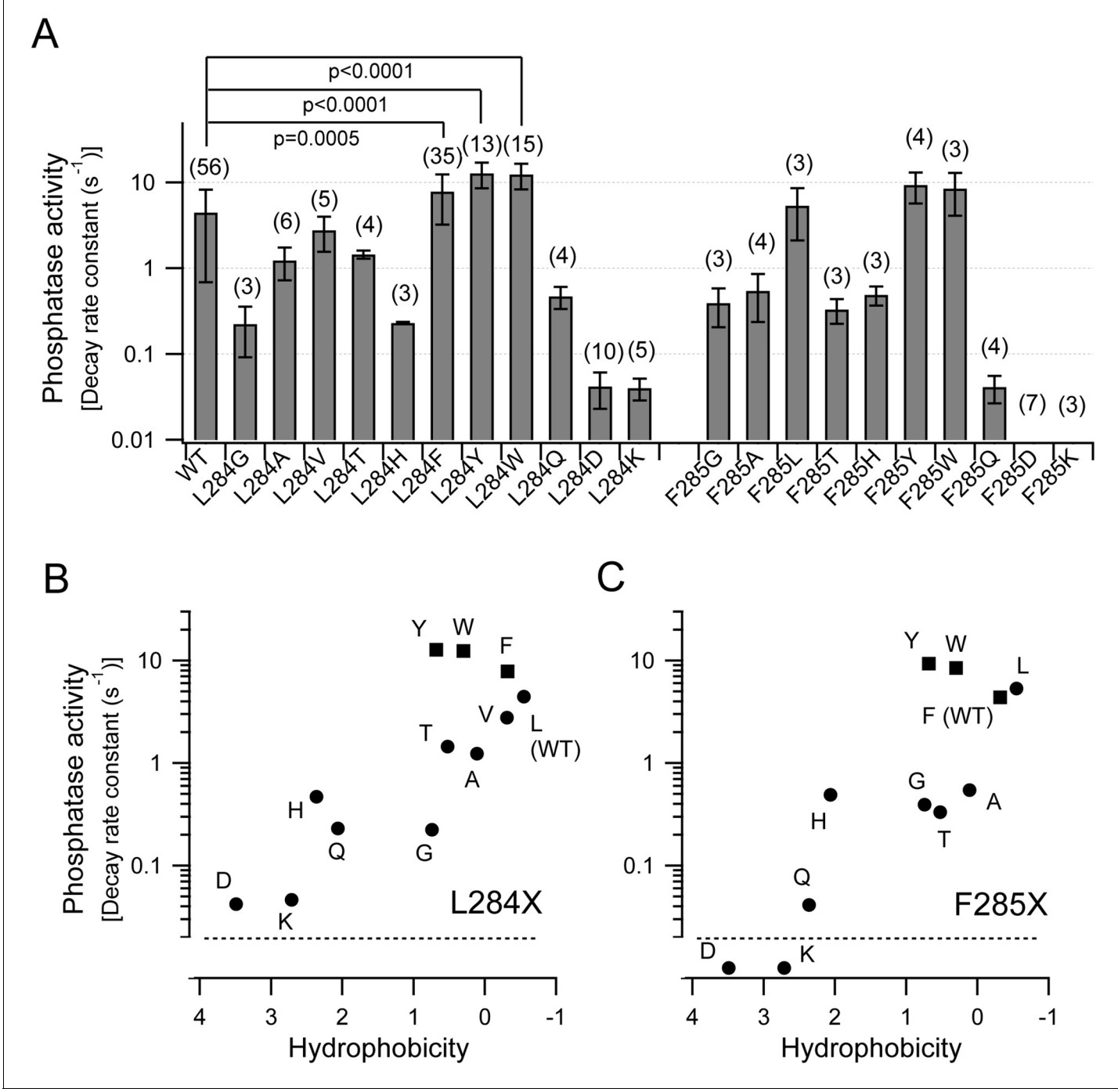

**Figure 5.** Voltage-dependent phosphatase activities of Ci-VSP depend on the hydrophobicity of the side chains of Leu-284 and Phe-285. (**A**) The phosphatase activities of Ci-VSP with mutations in the hydrophobic spine. Decay rate constants of the normalized Kir current (*Figure 4B* and *Figure 4—figure supplement 2*) calculated by single exponential fitting are shown as phosphatase activities. Data are the mean ± S.D. from 3 to 56 oocytes. Values in brackets indicate the number of oocytes. Upper bars show p-values from a two-tailed Student's *t*-test. (**B and C**) Phosphatase activities of Ci-VSP mutants at Leu-284 (**B**) or Phe-285 (**C**) plotted against the hydrophobicity of the amino acid side chains. The values of voltage-dependent phosphatase activities are from A. Hydrophobicity values are from *Hessa et al. (2005)*.

DOI: https://doi.org/10.7554/eLife.41653.016

The following source data and figure supplements are available for figure 5:

**Source data 1.** Numerical data that were used to generate the chart in *Figure 5*.
DOI: https://doi.org/10.7554/eLife.41653.020

*Figure 5 continued on next page*

*Figure 5 continued*

**Figure supplement 1.** Cell surface expression levels of Ci-VSP constructs as examined by sensing charges.
DOI: https://doi.org/10.7554/eLife.41653.017
**Figure supplement 2.** Little correlation between the voltage-dependent phosphatase activity and the side chain volume.
DOI: https://doi.org/10.7554/eLife.41653.018
**Figure supplement 3.** Voltage dependent phosphatase activities of Ci-VSP mutants examined with voltage-gated K$^+$ channel, Kv7.2/7.3.
DOI: https://doi.org/10.7554/eLife.41653.019

equal to 15.4 s$^{-1}$). To estimate the voltage-dependent phosphatase activity of WT, L284F, L284Y, and L284W more accurately, a shorter (50 ms) depolarization pulse protocol was utilized in another set of experiments (*Figure 6A*). The normalized Kir currents at the end of a test pulse were plotted as a function of the accumulated depolarization time (e.g., 1 st: 0 ms; 2nd: 50 ms) (*Figure 6B*). Decay time constants were fitted by a single exponential function, and surface expression levels of Ci-VSP mutants were determined by measuring the off-sensing charge (*Figure 6C and D*). Experiments using a shorter depolarizing pulse protocol verified that the voltage-dependent enzyme activity of mutants with an aromatic residue at position 284 was enhanced.

We also assessed the voltage-dependent phosphatase activities of Ci-VSP mutants using a PI(4,5) P$_2$-sensitive voltage-gated K$^+$ channel [rat-Kv7.2/7.3 (KCNQ2/3)]. Representative traces of outward K$^+$ currents with Ci-VSP mutants are shown in *Figure 5—figure supplement 3* using a protocol similar to that of the Kir channel. With WT Ci-VSP, an outward KCNQ2/3 current was detected in the first trace during a test pulse at 50 mV, and the following traces clearly showed decreased current amplitudes. Decay time constants of KCNQ2/3 currents were estimated by fitting a single exponential function, and expression levels were confirmed by measuring the off-sensing charge (*Figure 5—figure supplement 3*). A slow decay was observed with the hydrophilic mutant L284Q. A representative mutant with an aromatic amino acid, L284F, showed faster decay KCNQ2/3 current kinetics than the WT (*Figure 5—figure supplement 3*). These results are consistent with the results of the Kir channel experiments, confirming that the presence of the aromatic ring in the hydrophobic spine facilitates the voltage-dependent phosphatase activity of Ci-VSP.

## Hydrophobicity, not the aromatic ring, of the hydrophobic spine is essential for the enzyme activity of the isolated CCR

The hydrophobic spine likely contributes to the loose association of the PD with the membrane and movability of the PD in the membrane/water interface as suggested above by our simulation analysis. The hydrophobic spine is also conserved in PTEN, which does not contain a VSD. This raises the possibility that the hydrophobic spine alone is critical for enzymatic activity. To test this possibility, a malachite green assay was performed with the isolated CCR of WT Ci-VSP and hydrophobic spine mutants (residues 248–576: the initial sequence of the expressed Ci-VSP is shown in *Figure 7A*). We examined three mutants: L284F, with high voltage-dependent enzymatic activity, and L284Q and F285Q, both with low voltage-dependent enzymatic activity. We also measured the phosphatase activities of full-length human PTEN with mutations in the sites corresponding to Leu-284 and Phe-285 and compared them with those of non-mutated PTEN. To estimate the effects of mutations in the hydrophobic spine, we determined relative activities normalized to WT activity (*Figure 7B*). The differences in phosphatase activities of Ci-VSP WT, L284F and F285Q showed a similar spectrum as those of their counterparts of PTEN (hPTEN WT, V45F and Y46Q). However, mutation of V45Q in hPTEN and L284Q in Ci-VSP showed different effects on the phosphatase activities.

L284Q and F285Q showed low phosphatase activities in an in vitro assay, which is consistent with the voltage-dependent phosphatase activity shown by the Kir current in *Xenopus* oocytes, suggesting that hydrophobicity in the hydrophobic spine is important for the innate phosphatase activity of the CCR. On the other hand, we noted some discrepancy between the phosphatase activity of the CCR and the voltage-dependent enzymatic activity shown by the Kir current: the phosphatase activity of the isolated CCR of L284F was not greater than that of the WT, whereas the voltage-dependent phosphatase activity of the L284F mutant, as well as that of the other aromatic ring residues, was greater than that of the WT Ci-VSP (*Figures 5* and *6*). This indicates that effects of mutations in the hydrophobic spine on the voltage-dependent phosphatase activities cannot be explained simply by changes in the intrinsic phosphatase activity of the CCR.

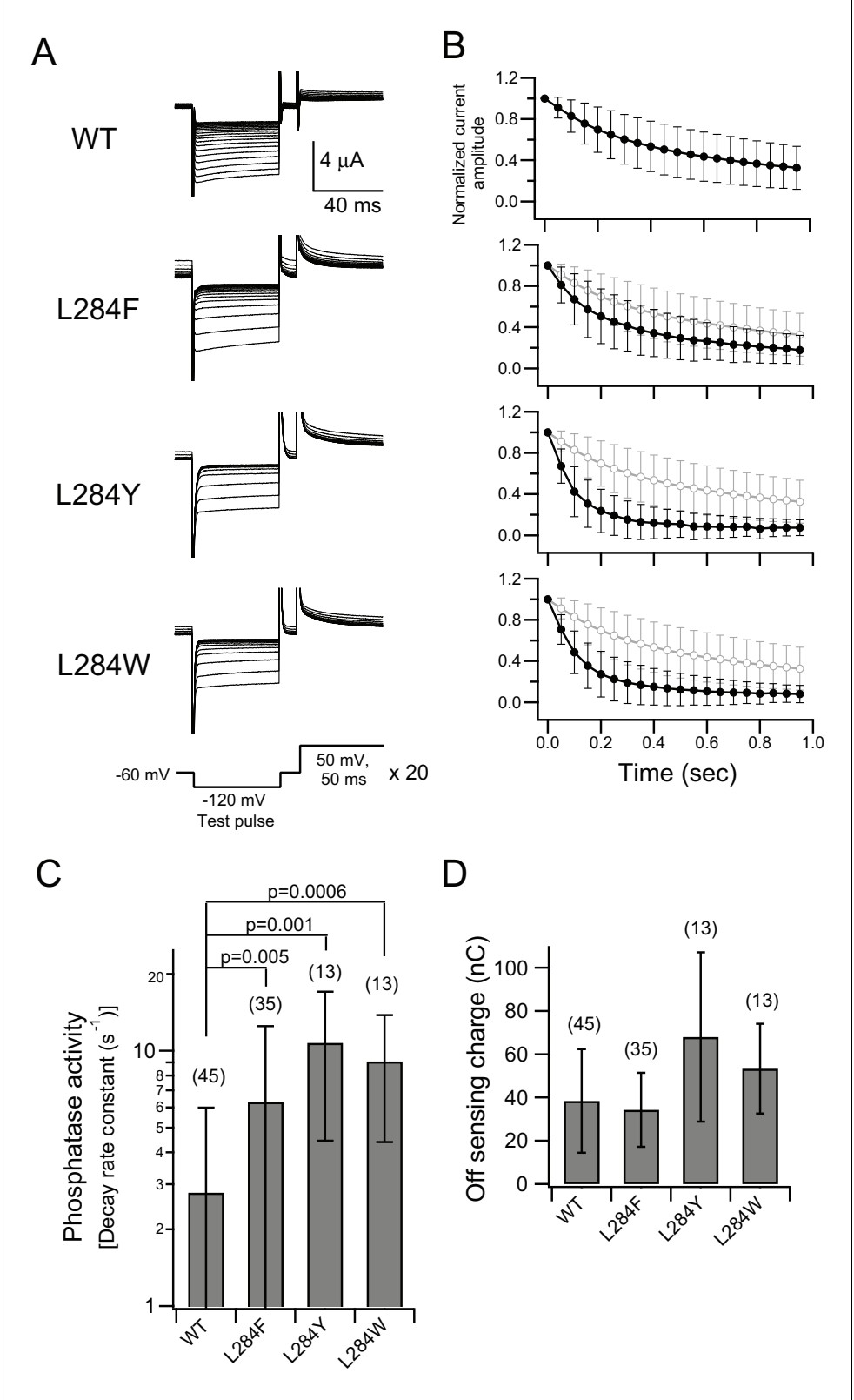

**Figure 6.** Addition of an aromatic ring to the hydrophobic spine enhances voltage-dependent enzymatic activity. (**A**) Representative current traces of a PI(4,5)P$_2$-sensitive K$^+$ channel (Kir3.2d, GIRK2) coexpressed with Ci-VSP WT, L284F, L284Y, and L284W in *Xenopus* oocytes. Pulse protocols (bottom panel) are a 50 ms test pulse and then 50 ms depolarization pulses (50 mV) repeated 20 times. All current traces are superimposed. The holding potential
*Figure 6 continued on next page*

*Figure 6 continued*

was maintained at −60 mV. Kir currents were observed during the test pulses, and the current amplitude indicated the relative amount of PI(4,5)P$_2$ in the membrane. (**B**) Plots of time-dependent normalized current amplitudes of the inward K$^+$ currents at the end point of the test pulse. Values were normalized to the first trace current amplitude of the test pulse. Black lines show the plots from current traces of WT, L284F, L284Y, and L284W. Gray lines show the plot from the WT, which is the same trace as that shown at the top. Symbols show the mean ± S.D. of data from 13 to 45 different oocytes (WT: n = 45, L284F: n = 35, L284Y: n = 13, L284W: n = 13). (**C**) The phosphatase activities of Ci-VSP mutated with aromatic amino acids at Leu-284. Decay rate constants of the normalized Kir current calculated by single exponential fitting are shown as phosphatase activities. Data are the mean ± S.D. from 13 to 45 oocytes. Values in brackets indicate the number of measured oocytes. Upper bars show p-values from a two-tailed Student's *t*-test. (**D**) Off-sensing charges of Ci-VSP mutants in each experiment corresponding to C, indicating expression levels at the cell surface. Sensing currents were measured by a repolarization step from 150 mV to holding potential (−60 mV), with a P/4–8 protocol used for subtracting leak currents and symmetrical capacitive currents. The integrated observed transient current indicates off-sensing charge. Data are the mean ± S.D. from 13 to 45 oocytes.
DOI: https://doi.org/10.7554/eLife.41653.021

The following source data is available for figure 6:

**Source data 1.** Numerical data that were used to generate the chart in *Figure 6B,C,D*.
DOI: https://doi.org/10.7554/eLife.41653.022

## Retrograde effects of mutations in the hydrophobic spine on VSD motion

VSD motion is known to be altered in a retrograde manner when VSD-CCR coupling is disrupted by the introduction of a mutation into the VSD-PD linker (*Kohout et al., 2010*). To understand how coupling of the VSD to the CCR is changed in Ci-VSP constructs with mutations in the hydrophobic spine, we examined VSD motion using a voltage clamp fluorometry (VCF) method (*Mannuzzu et al., 1996*), as previously reported (*Kohout et al., 2008*; *Sakata and Okamura, 2014*). The fluorescent probe tetramethylrhodamine-maleimide (TMRM) was attached to the introduced cysteine, which replaced Gly-214 on the S3-S4 extracellular loop.

Fluorescence changes upon membrane depolarization were detected in all of the mutants with G214C-TMRM (*Figure 8A*). Voltage-dependent fluorescence decreases of the WT and mutants were observed previously (*Kohout et al., 2008*). The averages of the normalized fluorescence changes during the last 1 ms (black arrow head in *Figure 8A*) of 500 ms test pulses were plotted against the membrane potential (*Figure 8B*). The normalized F-V relationship of L284F/G214C-TMRM shows that the fluorescence saturated at 150 mV as in WT/G214C-TMRM, whereas it did not saturate even at 150 mV in L284Q/G214C-TMRM and F285Q/G214C-TMRM. These results were similar to those of the charge-voltage (Q-V) relationships (*Figure 8—figure supplements 1* and *2*).

We compared the kinetics of the fluorescence changes of the WT and mutants of the hydrophobic spine with TMRM at the site of Gly-214 during the depolarization phase. *Figure 8—figure supplement 3* shows the fluorescence change of L284F/G214C-TMRM or L284Q/G214C-TMRM superimposed on that of WT/G214C-TMRM. At membrane voltages less than 50 mV, all three constructs showed similar kinetics: the times to half-maximum fluorescence change (t$_{1/2}$) of the three constructs were similar (*Figure 8—figure supplement 3C*). On the other hand, the kinetics of the fluorescence changes of the three constructs were different at voltages > 50 mV (t$_{1/2}$: L284Q < WT < L284F) (*Figure 8—figure supplement 3C*). The time-dependent fluorescence changes were also fitted by single exponential functions at voltages < 50 mV but by double exponentials at voltages > 50 mV (*Figure 8—figure supplement 4A*). The slow component proportion was larger in the L284F mutant than in the WT but smaller in L284Q. Upon membrane repolarization, the mutants showed different kinetics of fluorescence change (*Figure 8—figure supplements 3D* and *4B*), which is consistent with the decay kinetics of off-sensing currents (*Figure 8—figure supplement 4C*).

We next analyzed the fluorescence changes of a tetramethylrodamine-maleimide (TMRM) at another position in the VSD (Q208C-TMRM). The averages of the normalized fluorescence changes of Q208C-TMRM at the time point of peak signal of the 200 mV trace from WT (black arrow heads in *Figure 8—figure supplement 5A*) were plotted against the membrane potential (F-V relationship; black filled circles, *Figure 8—figure supplement 5B*). As previously reported, complex time-

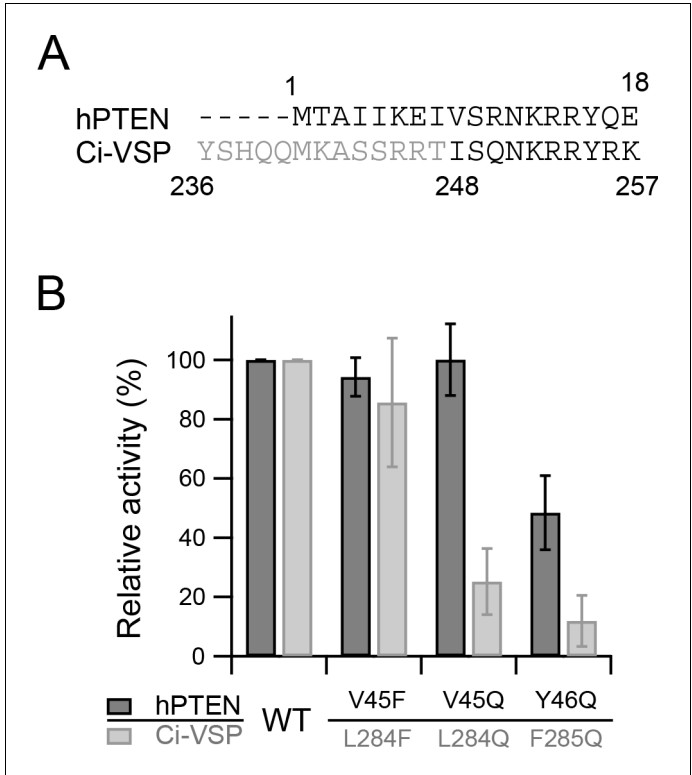

**Figure 7.** In vitro phosphatase activity of the isolated cytoplasmic catalytic region of Ci-VSP. (**A**) Amino acid sequences of hPTEN and the initiating region of the phosphatase domain of Ci-VSP. The regions of proteins used in the in vitro phosphatase activity assay are shown in black letters (full-length PTEN and the cytoplasmic catalytic region of Ci-VSP [248–576]). (**B**) Relative phosphatase activities of recombinant hPTEN mutants (WT, V45F, V45Q, and Y46Q) and Ci-VSP mutants (WT, L284F, L284Q, and F285Q) determined by a malachite green assay with di-$C_{16}$-PI(3,4,5)$P_3$ and di-$C_{16}$-PI(4,5)$P_2$, respectively. Error bars indicate S.D. (hPTEN: n = 6, Ci-VSP: n = 9). The values are normalized to the amount of released phosphate by hPTEN WT or Ci-VSP WT, respectively.
DOI: https://doi.org/10.7554/eLife.41653.023

The following source data is available for figure 7:

**Source data 1.** Numerical data that were used to generate the chart in *Figure 7B*.
DOI: https://doi.org/10.7554/eLife.41653.024

dependent fluorescence changes were observed (*Kohout et al., 2008*; *Kohout et al., 2010*; *Grimm and Isacoff, 2016*). The F-V relationship of Ci-VSP Q208C-TMRM showed that fluorescence changes have three phases along the voltage axis. There are no significant differences among WT and mutants in the first and second phases of the F-V relationship observed at voltages < 100 mV. WT and L284F/Q208C-TMRM had a third component, but the other mutants did not. Previously, the third component was shown to correlate with VSD-CCR coupling, based on an analysis of the decoupled constructs with VSD-PD linker mutations (*Kohout et al., 2010*).

Therefore, our results indicate that VSD motion was affected by the behavior of the hydrophobic spine.

## Analysis of the fluorescence of an unnatural amino acid, Anap, suggests two-step activation of the CCR

We previously studied the conformational changes of the CCR of Ci-VSP by analyzing the fluorescence changes of a genetically encoded unnatural fluorescent amino acid, 3-(6-acetylnaphthalen- 2-ylamino)−2-aminopropanoic acid (Anap) in *Xenopus* oocytes (*Sakata et al., 2016*). The method enabled us to estimate conformational changes in living cells with minimal perturbation of the protein structure because Anap has a size similar to that of natural amino acids. Using this method, the

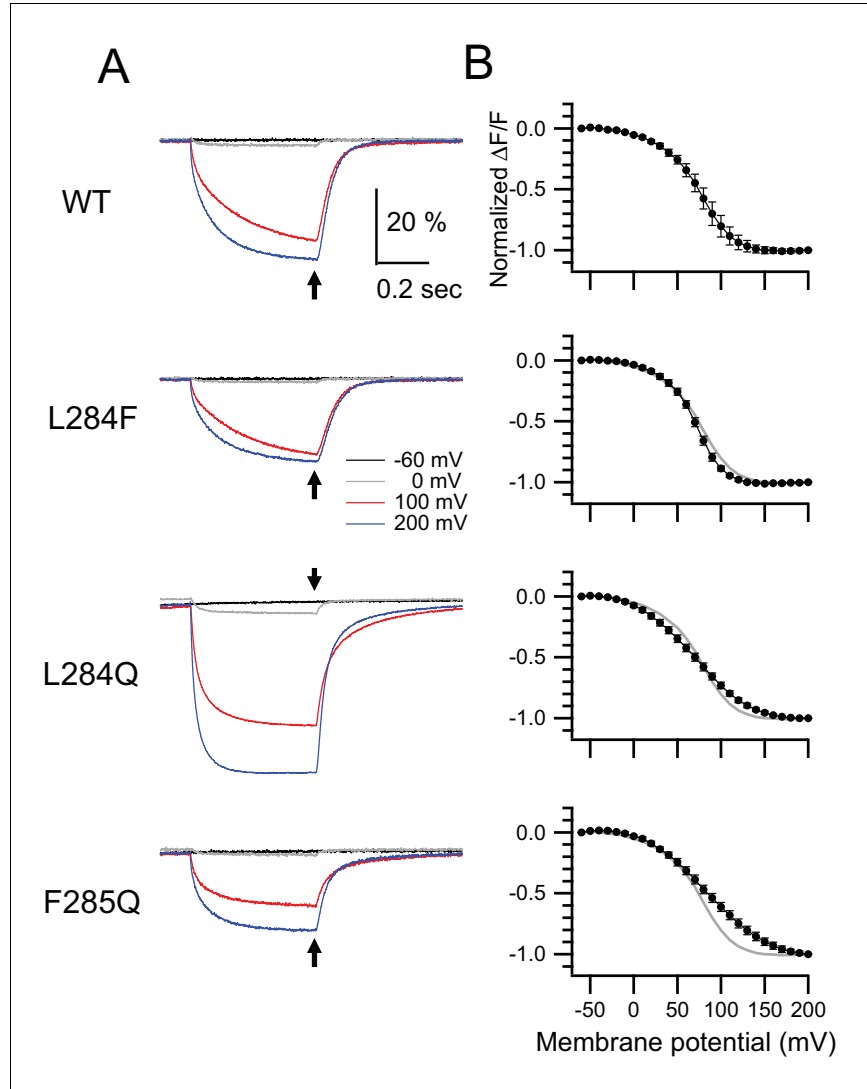

**Figure 8.** Retrograde effects on voltage sensor domain motion by mutations in the hydrophobic spine. Voltage-clamp fluorometry of Ci-VSP and its hydrophobic spine mutants labeled by TMRM at Gly-214 on the S3-S4 linker of the voltage sensor domain. (**A**) Representative fluorescence changes of Ci-VSP WT, L284F, L284Q, and F285Q, with G214C-TMRM measured using 500 ms depolarizing pulses. Black, gray, red, and blue lines show traces at −60, 0, 100, and 200 mV, respectively. The holding potential was maintained at −60 mV. Traces are normalized to the fluorescence level before depolarization. (**B**) Normalized fluorescence-voltage (**F–V**) relationship of Ci-VSP WT and mutants. The values at arrows in A were normalized to that at 200 mV and plotted. Error bars indicate S.D. (G214C-TMRM: WT: n = 6, L284F: n = 6, L284Q: n = 6, F285Q: n = 6).

DOI: https://doi.org/10.7554/eLife.41653.025

The following source data and figure supplements are available for figure 8:

**Source data 1.** Numerical data that were used to generate the chart in *Figure 8B*.
DOI: https://doi.org/10.7554/eLife.41653.031
**Figure supplement 1.** Charge-voltage relathionships of the voltage sensor domain.
DOI: https://doi.org/10.7554/eLife.41653.026
**Figure supplement 2.** The fluoroscence-voltage relationships versus the charge-voltage relationships.
DOI: https://doi.org/10.7554/eLife.41653.027
**Figure supplement 3.** Kinetic analysis of the voltage sensor motion by VCF (1).
DOI: https://doi.org/10.7554/eLife.41653.028
**Figure supplement 4.** Kinetic analysis of the voltage sensor motion by VCF (2).
DOI: https://doi.org/10.7554/eLife.41653.029

*Figure 8 continued on next page*

*Figure 8 continued*

**Figure supplement 5.** Retrograde effects on voltage sensor motion by mutations with Q208C-TMRM on the S3-S4 linker.

DOI: https://doi.org/10.7554/eLife.41653.030

motion of the CCR in the hydrophobic spine mutants was examined. Anap was incorporated at Lys-555 in the membrane-facing side of the C2 domain, and VCF was performed (*Sakata et al., 2016*).

*Figure 9* shows representative traces of Anap fluorescence from the WT and hydrophobic spine mutants. As previously reported (*Sakata et al., 2016*), we found that Ci-VSP K555Anap exhibited two characteristic fluorescence changes, a small early decrease at low membrane voltages and a large late increase at higher membrane voltages. Small decreases in fluorescence upon membrane depolarization at over 20 mV were observed. At more than 80 mV, the voltage-dependent fluorescence changes were biphasic: fast fluorescence decreases were detected, similar to those at low membrane voltages, followed by large slow increases, larger amplitudes, and faster kinetics as the membrane voltage increased (*Figure 9*; $t_{1/2}$ and time constants are shown in *Figure 9—figure supplement 1*). The later increases were more remarkable in L284F/K555Anap (*Figure 9A*), and the F-V relationships showed a larger fluorescence increase from L284F/K555Anap than from WT/K555Anap at higher membrane potentials, with some leftward shift (*Figure 9B* and *Figure 9—figure supplement 2*). Analysis of F-V relationships revealed that both the amplitude increase and leftward shift of voltage dependence underlie the larger fluorescence increase of the second component (*Figure 10B* and *Table 2*). In contrast, L284Q/K555Anap and F285Q/K555Anap did not show the second fluorescence change, exhibiting only a small decrease in fluorescence even at high membrane voltage, which is similar to the earlier fast decrease in WT/K555Anap (*Figure 9*: $t_{1/2}$ values are shown in *Figure 9—figure supplement 1*). Surface expression was confirmed by off-sensing charges (*Figure 9C*).

To gain more insight into the relationship between VSD motion and CCR conformational changes, we compared F-V curves of two different fluorescent probes (G214C-TMRM for the VSD and K555Anap for the CCR) in several constructs with mutations in the hydrophobic spine. F-V curves were fitted by one or the sum of two Boltzmann distributions (*Figure 10* and *Figure 10—figure supplement 1*). For both G214C-TMRM and K555Anap, F-V curves of WT and L284F were fit well by the sum of two Boltzmann distributions, indicating that they had two individual components (red and blue lines in *Figure 10* and *Figure 10—figure supplement 1*; fitting parameters are shown in *Table 2*). The voltages of half-maximum changes ($V_{1/2}$) were identical to each other in the early components of WT/G214C-TMRM and WT/K555Anap, as well as L284F/G214C-TMRM and L284F/K555Anap (*Table 2*). Interestingly, later components were only observed in the WT and L284F, both with fluorescence probes (*Table 2*), providing strong evidence for tight coupling between the VSD and CCR in the WT and L284F. On the other hand, a Boltzmann distribution was sufficient to fit the F-V curves of L284Q and F285Q with both fluorescence probes (*Table 2*). Signal amplitudes of fluorescence from L284Q/K555Anap and F285Q/K555Anap were similar to those of early components of WT/K555Anap and L284F/K555Anap (A versus $A_1$ in *Table 2*), suggesting that the early components of WT and L284F share molecular events with the conformational change of the CCR in L284Q and F285Q.

These findings indicate that the CCR is activated through two steps that are tightly coupled with the VSD activation state. The second activation step is more closely associated with enzyme activity, in which the hydrophobic spine plays a role.

## Discussion

In the present study, through computational and experimental approaches, we identified a novel membrane-interacting site in the cytoplasmic catalytic region (CCR) of the voltage-sensing phosphatase (VSP), which plays an important role in coupling of the voltage sensor domain (VSD) to the CCR. Our MD simulations of the CCR suggested that the combination of the hydrophobic spine, consisting of two conserved hydrophobic amino acids (Leu-284 and Phe-285 in Ci-VSP), and the flanking hydrophilic residues provide high mobility to the PD in and near the membrane/water interface. Electrophysiological studies of constructs with systematic mutations in the hydrophobic spine

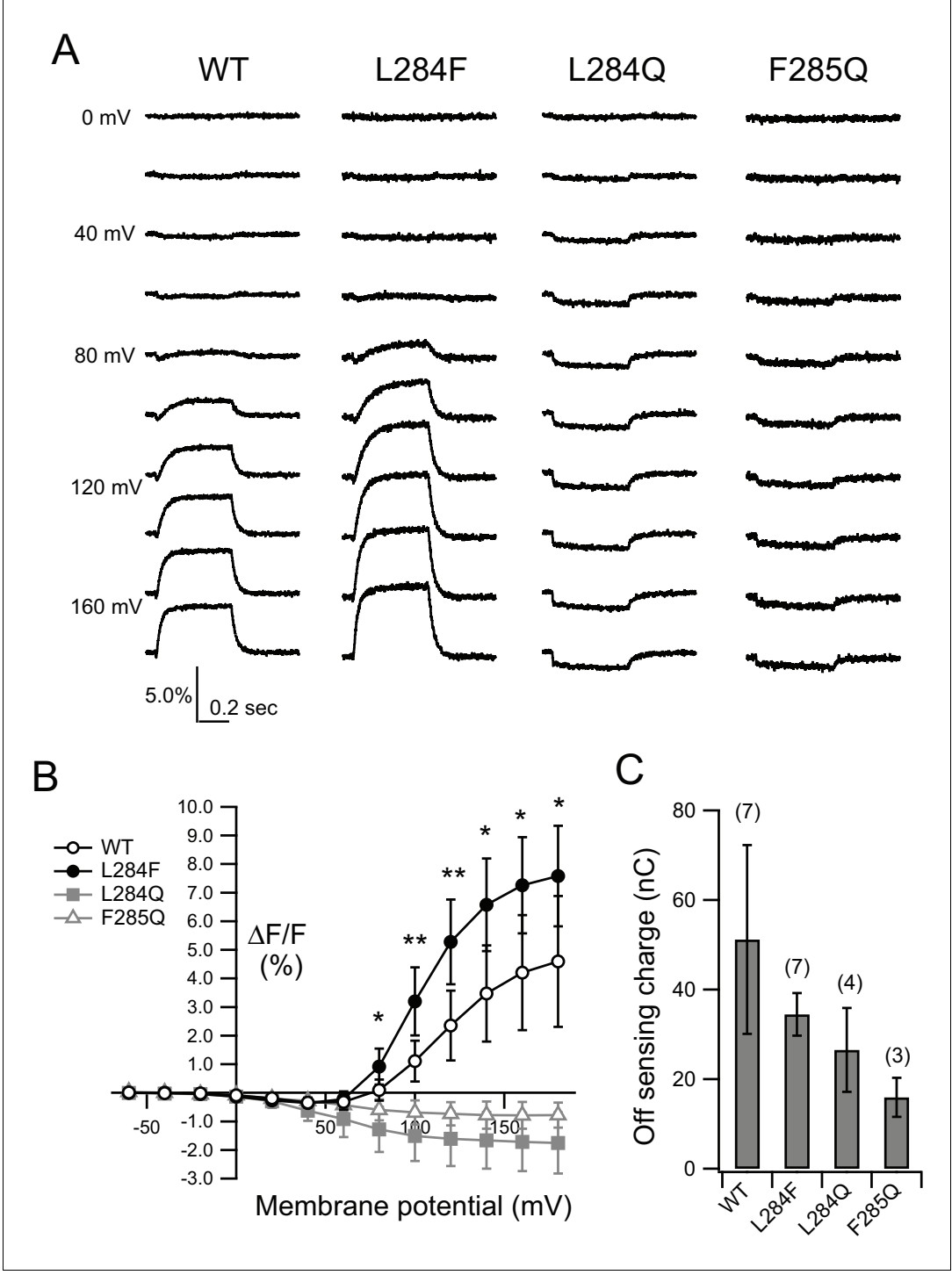

**Figure 9.** Detection of voltage-dependent conformational changes in the cytoplasmic catalytic region of VSP using an unnatural fluorescent amino acid, Anap. (**A**) Representative voltage-dependent fluorescence changes of an unnatural amino acid, Anap, incorporated at Lys-555 with mutations in the hydrophobic spine (WT, L284F, L284Q, and F285Q), measured with 500 ms depolarizing pulses ranging from 0 to 180 mV in 20 mV increments. The holding potential was maintained at −60 mV. Traces are normalized to the fluorescence level before depolarization. (**B**) Fluorescence-voltage (F–V) relationships of Ci-VSP WT and mutants. Values at the end point of test pulse in A were plotted. Black blank circles, black circles, gray squares, and gray blank triangles show the fluorescence changes of WT, L284F, L284Q, and F285Q, respectively. Error bars indicate S.D. (WT: n = 7, L284F: n = 7, L284Q: n = 4, F285Q: n = 3). The statistical significance of differences between WT and L284F were

*Figure 9 continued on next page*

*Figure 9 continued*
evaluated by a two-tailed Student's *t*-test (*: p<0.05, **: p<0.005). (C) Sensing charges calculated from the off-sensing current measured following the 160 mV depolarizing step. The data were collected from the same oocytes described in B. Error bars indicate S.D., and the brackets show the number of experiments.
DOI: https://doi.org/10.7554/eLife.41653.032
The following source data and figure supplements are available for figure 9:

**Source data 1.** Numerical data that were used to generate the chart in *Figure 9B,C*.
DOI: https://doi.org/10.7554/eLife.41653.035
**Figure supplement 1.** Kinetic analysis of the cytoplasmic conformational changes as reported by the fluorescence of K555Anap.
DOI: https://doi.org/10.7554/eLife.41653.033
**Figure supplement 2.** Addition of an aromatic residue to the hydrophobic spine caused leftward shift of the F-V relationship of Anap introduced into the C2 domain.
DOI: https://doi.org/10.7554/eLife.41653.034

showed that voltage-dependent phosphatase activity depends on the hydrophobicity of the amino acid side chain. Interestingly, the addition of extra aromatic ring residues significantly enhanced this phosphatase activity. Voltage clamp fluorometry showed that both hydrophobicity and the presence of an aromatic ring on the hydrophobic spine controlled the later state transition of the two sequential CCR activation steps. We propose that the hydrophobic spine plays a critical role in regulating coupling of the VSD to dynamic rearrangements in the CCR for enzymatic activity.

## The hydrophobic spine controls the later transition of enzyme activation

Previous studies have suggested the presence of multiple activation states of enzymes that are coupled with distinct states of the VSD (*Kohout et al., 2010*; *Sakata and Okamura, 2014*; *Grimm and Isacoff, 2016*; *Sakata et al., 2016*). However, detailed mechanisms of the multiple conformations of the CCR upon VSD activation remain unclear.

The biphasic fluorescence changes of Anap incorporated into the CCR (K555Anap) with a mutation in the hydrophobic spine were analyzed (*Figure 9*, *Figure 10B* and *Table 2*). The second slowly increasing component of fluorescence at higher membrane voltages was absent in the mutants L284Q and F285Q, which showed lower voltage-dependent phosphatase activity. On the other hand, the second phase of the fluorescence change was enhanced in the mutant L284F, which showed strong voltage-dependent phosphatase activity. To gain more insight, we also measured another two mutants in the hydrophobic spine (L284T for a mutation with a moderate enzyme activity and L284K for a mutation with a low enzyme activity) with K555Anap (*Figure 10—figure supplement 2*). In the case of L284T, biphasic fluorescence changes of Anap were observed. Interestingly, the later increase was significantly smaller than that of WT, which is strong evidence for the later component as the conformation of a high enzymatic activity. On the other hand, L284K showed only a small fluorescent decrease, as L284Q. These findings indicate that VSP has two-active states with different CCR activities and that the hydrophobicity and presence of an aromatic ring in the hydrophobic spine drive the state transition to a fully activated state, thus ensuring robust enzyme activity.

These ideas were also supported by retrograde effects of mutation of the hydrophobic spine on the VSD, which was examined in a VCF experiment. The fluorescence changes of TMRM introduced in Gly-214, an extracellularly facing movable site of the VSD, demonstrated similar effects of mutation to those of Ci-VSP K555Anap (*Figure 10*). F-V curves were fitted by the sum of two Boltzmann distributions in the WT and L284F, which had distinct voltage dependence, and the voltage dependence of the second component was shifted leftward in L284F as in the case for K555Anap, whereas L284Q and F285Q showed curves fitted by only one Boltzmann distribution. These results indicate that the transition states of the VSD are linked with those of the CCR, consistent with the idea that a mutation in the hydrophobic spine influences the transition between the two-active states of the CCR. To summarize, there are two CCR active states and the transition between the two active states requires tight coupling between the VSD and CCR that is dependent on the nature of the hydrophobic spine (*Figure 11*).

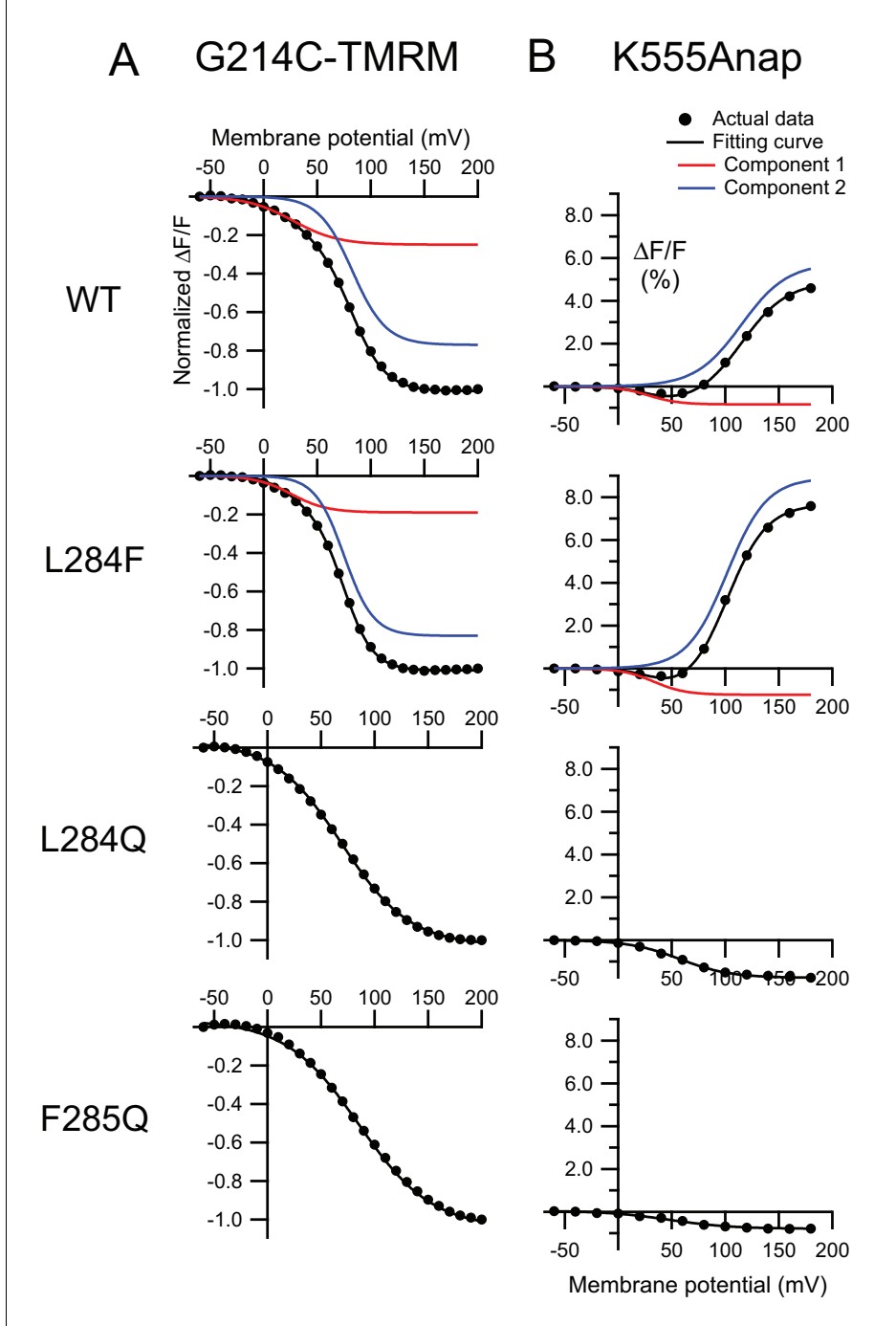

**Figure 10.** Voltage-dependent conformational changes in the VSD tightly link to those in the CCR. Fluorescence-voltage relationships and fitting results of the Ci-VSP WT and mutants with different fluorescent probes [G214C-TMRM (**A**) or K555Anap (**B**) ]. F-V relationships from experimental data in *Figure 8B* or 9B (black circles) were fitted with one or the sum of two Boltzmann distributions (black solid line). Red and blue lines show individual two components fitted with the sum of two Boltzmann distributions. All fitting parameters and equations are shown in *Table 2* and its legend.

DOI: https://doi.org/10.7554/eLife.41653.036

The following source data and figure supplements are available for figure 10:

**Source data 1.** Numerical data that were used to generate the chart in *Figure 10*.
DOI: https://doi.org/10.7554/eLife.41653.039

**Figure supplement 1.** Fitting the F-V relationships with single or double Boltzmann distributions.

*Figure 10 continued on next page*

*Figure 10 continued*

DOI: https://doi.org/10.7554/eLife.41653.037

**Figure supplement 2.** Evaluation of voltage-dependent cytoplasmic conformational changes in another two mutations at the hydrophobic spine using K555Anap .

DOI: https://doi.org/10.7554/eLife.41653.038

The positively charged residues in the VSD-PD linker have previously been suggested to be important for coupling between the two domains (*Kohout et al., 2010*). The fluorescence changes of TMRM introduced into another site (Gln-208) in weak enzyme activity mutants (L284Q and F285Q, *Figure 8—figure supplement 5*) showed similar voltage-dependent changes in fluorescence as those of Ci-VSP with mutations that affected the positive charges in the VSD-PD linker (K252Q, R253Q) in which the VSD is completely decoupled from the CCR (*Kohout et al., 2010*). These results suggested that the VSD-PD linker is required to couple VSD motion to the CCR, which initiates VSP enzyme activity. The hydrophobic spine also affects VSD-CCR coupling, but it has a different role than that of the VSD-PD linker. This spine mainly affects the later transition of the CCR, suggesting that it is needed to promote the state transition from the intermediate state to the fully activated state, whereas interaction between the VSD-PD linker and the active center of the PD is indispensable for initiating all enzymatic activities.

## Mechanistic insights into coupling between the VSD and CCR

The CG and AT MD simulations of the isolated CCR in this study show that the C2 domain is anchored to the plasma membrane through positive charges at the membrane interface of the C2

**Table 2.** Fitting parameters of fluorescence-voltage relationships of Ci-VSP G214C-TMRM and K555Anap.

F-V curves (Normalized $\Delta F/F$ or $\Delta F/F$ vs voltage) shown in *Figure 10* were fitted by single or two components of Boltzmann distribution, $F(V)=A/[1 + \exp((V-V_{1/2})/slope]$ or $F(V) = A_1/[1 + \exp((V-V_{1/2,1})/slope_1] + A_2/[1 + \exp((V-V_{1/2,2})/slope_2]$. Where, 'A' is the amplitude of each component, 'V' is the membrane potential for test pulse, '$V_{1/2}$' is the half-maximum potential, 'slope' is the steepness of the curve. In the case of G214C-TMRM, 'Ratio' in this table is calculated by $A_1/(A_1+A_2)$ or $A_2/(A_1+A_2)$. In K555Anap, 'A' in this table indicates the absolute fluorescence change of each component. All data are shown as mean ± S.D. The statistical significance of differences were evaluated by a two-tailed Student's t-test

| G214C-TMRM | | Ratio (%) | $V_{1/2}$ (mV) | N |
|---|---|---|---|---|
| WT | component 1 | 24.3 ± 7.0 | 27.1 ± 4.3 | 6 |
| | component 2 | 75.7 ± 7.0 | 82.7 ± 5.5[a] | |
| L284F | component 1 | 18.6 ± 5.7 | 23.4 ± 6.6 | 6 |
| | component 2 | 81.4 ± 5.7 | 75.1 ± 1.2[a] | |
| L284Q | | - | 69.2 ± 4.1 | 6 |
| F285Q | | - | 85.1 ± 4.4 | 6 |
| | | | | |
| **K555Anap** | | A (%) | $V_{1/2}$ (mV) | N |
| WT | component 1 | −0.85 ± 0.48 | 28.3 ± 10.1 | 7 |
| | component 2 | 5.76 ± 2.81[b] | 115.0 ±11.3[c] | |
| L284F | component 1 | −1.23 ± 0.34 | 33.1 ± 4.3 | 7 |
| | component 2 | 8.90 ± 1.92[b] | 101.6 ±3.9[c] | |
| L284Q | | −1.76 ± 1.08 | 57.2 ± 6.76 | 4 |
| F285Q | | −0.84 ± 0.46 | 47.3 ± 14.8 | 3 |

a, p=0.026; b, p=0.046; c, p=0.028 by two-tailed Student's t-test.

DOI: https://doi.org/10.7554/eLife.41653.040

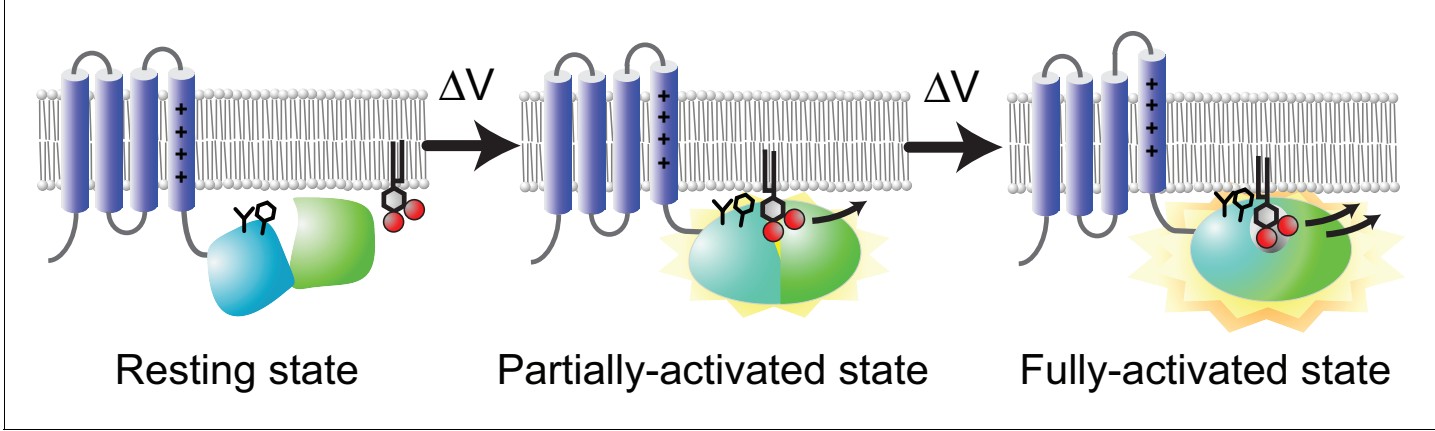

**Figure 11.** Schematic illustration of VSP activity: A model of two-step activation. Two-step conformational and activity changes in VSP. The black sticks on the PD shows the hydrophobic spine consisting of amino acid side chains of Leu-284 and Phe-285. In the resting state (left), the PD (turquoise) forms no or weak interactions with the plasma membrane, which cannot catalyze the substrate. At low membrane voltages (middle), the PD is pulled to the membrane by the partially activated VSD (blue), which shows low phosphatase activity. At higher membrane voltages, the VSD is fully activated, and the PD is tightly anchored to the membrane surface, inducing high phosphatase activity. The weak enzyme activity at the middle state is consistent with the results of the Anap study of L284Q and L284K mutants which showed low enzyme activity (*Figure 10—figure supplement 2*).
DOI: https://doi.org/10.7554/eLife.41653.041

The following figure supplement is available for figure 11:

**Figure supplement 1.** Membrane interface around PIPs binding region in various phosphoinositides phosphatases.
DOI: https://doi.org/10.7554/eLife.41653.042

domain (*Figure 2*). On the other hand, PD is more loosely bound to the membrane because of the hydrophobic nature of the hydrophobic spine, suggesting that phosphatase activity is regulated by the hinge-like motion of the CCR through membrane binding and unbinding of PD in the membrane-anchored C2 domain. The location of the hydrophobic residues in such a polar microenvironment (i.e., phosphate and choline groups and water) is an anomalous feature of the hydrophobic spine, given that the side chains of Leu and Phe favor interactions with the deep hydrophobic core of phospholipids, with an approximate free energy of −20 and −14 kJ/mol, as inferred from simulations and experiments (*MacCallum et al., 2007*; *MacCallum et al., 2008*; *Cordomí et al., 2012*). This result may seem to conflict with our previous results that the distance between the CCR and plasma membrane was not significantly changed upon voltage-induced activation, based on the measurement by FRET between Anap and a membrane-embedded FRET quencher, dipicrylamine (DPA) (*Sakata et al., 2016*). However, it should be noted that we estimated the change in distance between the membrane and the C2 domain but not the phosphatase domain. Alternatively, there may be a detection limit for distance change in the FRET analysis.

Our MD simulation analyses suggested that the hydrophobic spine, likely in combination with flanking hydrophilic amino acid residues, regulates the membrane-PD interaction by fine-tuning the balance of hydrophobicity/hydrophilicity in the region. This suggestion was confirmed by electrophysiological and in vitro studies of Ci-VSP mutants. An experiment with a series of mutations into the hydrophobic spine showed that hydrophobicity is essential for maintaining phosphatase activity. The addition of an aromatic ring in the hydrophobic spine greatly enhanced voltage-dependent phosphatase activity but showed no effect on the innate phosphatase activity of the CCR. Further analysis of voltage-dependent conformational changes showed that the aromatic ring of the hydrophobic spine affects the transition of CCR between two active states, suggesting that the aromatic ring may stabilize the high activity state through tight interactions with the plasma membrane. There are some examples in which aromatic amino acids help proteins insert themselves into membranes. Studies of the membrane-tethered $\beta2$ subunit of voltage-gated $Ca^{2+}$ channel ($Ca_V$) suggested that $Ca_V$ binds to the plasma membrane through electrostatic and hydrophobic interactions with basic and aromatic residues (*Miranda-Laferte et al., 2014*; *Kim et al., 2015*; *Kim and Suh, 2016*; *Park et al., 2017*). In a study of phospholipase $A_2$ ($PLA_2$), tryptophan plays an important role in binding to the plasma membrane, and removal of tryptophan at the interface lowers the affinity of $PLA_2$

to the membrane (*Gelb et al., 1999*). Subtle differences among hydrophobic amino acid side chains in their dynamics in and near the phospholipid head groups can have profound impacts on voltage-dependent phosphatase activity.

We propose a model (*Figure 11*) based on our results. When the VSD stays in the down state (the resting state), PD demonstrates no or weak interactions with the membrane phospholipids, so that the CCR cannot form a stable complex with substrates and no catalytic activity occurs. Under mild depolarization in which the VSD is partially activated, VSD motion is transmitted to the PD through the VSD-PD linker by pulling the PD up toward the membrane surface to form weak interactions with the membrane. At higher membrane voltages, the VSD is fully activated and the PD is tightly anchored to the membrane surface, inducing conformational changes in the CCR and a fully activated state with high phosphatase activity. In contrast, the C2 domain can tightly bind to membrane phospholipids at any time, supporting a fast response to membrane potential changes in VSP by keeping the PD close to the membrane.

In this model, we suggest that the hydrophobic spine in the VSP assists the tight coupling between the VSD and CCR by fine-tuning the membrane association of the PD. The hydrophobicity and aromatic ring in the hydrophobic spine are critical for this membrane association. Detailed mechanisms from a structural point of view will be discussed in the next section.

## The hydrophobic spine shows features that are common among phosphoinositide phosphatases

The amino acid sequence of the CCR is highly homologous to that of PTEN, and the two crystal structures are similar (*Matsuda et al., 2011*; *Liu et al., 2012*). The hydrophobic spine is also conserved in PTEN (*Figure 3A*), and its structure is directed toward membrane phospholipids, similar to VSP. Results of our in vitro malachite green assay showed that the hydrophobic spine affects the enzyme activity of PTEN (*Figure 7*). Crystal structures suggest that hydrophobic amino acids near the substrate binding site play important roles in substrate binding and recognition (*Hsu et al., 2012*; *Trésaugues et al., 2014*) (*Figure 11—figure supplement 1*). A *Legionella* PI 3-phosphatase, SidF, hydrolyzes the 3-phosphates of PI(3,4)$P_2$ and PI(3,4,5)$P_3$. In the crystal structure of SidF with diC$_4$-PI(3,4)$P_2$, Trp-420 and Phe-421 form hydrophobic protrusion loops that are oriented toward membrane lipid and show hydrophobic interactions with the acyl chain at the sn-2 position of diC$_4$-PI(3,4)$P_2$ in the substrate binding pocket (*Hsu et al., 2012*). Structures of the catalytic domains of proteins in the human 5-phosphatase family, INPP5B, OCRL, and SHIP2 also contain a highly hydrophobic cluster at the membrane surface (*Trésaugues et al., 2014*) (*Figure 11—figure supplement 1*). The INPP5B structure reveals that two hydrophobic clusters, LC1R (Phe-311, Phe-312, and Phe-313) and LC2R motifs (Ile-373, Met-374, and Met-377), interact with the acyl chain at the sn-2 position of the diC$_8$-PI(3,4)$P_2$ substrate. OCRL has similar hydrophobic structure for binding the acyl chain of substrates, whereas SHIP2 lacks the region corresponding to the LC1R motif in INPP5B, and an in vitro phosphatase activity assay showed an isolated catalytic domain of SHIP2 with very low activity compared to that in the INPP5B and OCRL catalytic domain (*Trésaugues et al., 2014*). These structures are necessary for interaction with the acyl chain of the phosphoinositide and are anchored to the plasma membrane, suggesting that its binding to phosphoinositides allows substrate recognition. Interestingly, all of the hydrophobic clusters containing aromatic residues interact with the sn-2 acyl chain, not with the sn-1 chain.

Therefore, a characteristic cluster formed by hydrophobic residues near the binding pocket seems to be a common structural feature of phosphoinositide phosphatases. The hydrophobic spine of VSP may interact with the acyl chain of the substrate, as found in INPP5B, OCRL, and SHIP2. From a structural point of view, the spine contributes to the formation of the stable substrate binding pocket with active center, CBR3 loop, and gating loop, because mutations in the hydrophobic spine affect innate phosphatase activity in the CCR. The substrate must be pulled from membrane phospholipids for dephosphorylation by VSP. The hydrophobic spine may help the hydrophobic acyl chains of the substrate stabilize in the hydrophilic cytoplasmic environment. A detailed understanding of the relationships between the VSD, the CCR, and the substrate await elucidation of the high resolution structure of the whole VSP protein with phosphoinositide substrates.

# Materials and methods

## Key resources table

| Reagent type (species) or resource | Designation | Source or reference | Identifiers | Additional information |
|---|---|---|---|---|
| Gene (Ci.intestinalis) | Ci-VSP | *Murata et al. (2005)* (PMID: 15902207) | GeneBank ID: NM_001033826 | |
| Recombinant DNA reagent | human-PTEN (plasmid) | | | gift from Dr. Tomohiko Maehama |
| Recombinant DNA reagent | mouse-Kir3.2d (plasmid) | | | gift fom Dr. Yoshihisa Kurachi |
| Recombinant DNA reagent | rat-Kv7.2 (plasmid) | | | gift from Drs. David McKinnon and Koichi Nakajo |
| Recombinant DNA reagent | rat-Kv7.3 (plasmid) | | | gift from Drs. David McKinnon and Koichi Nakajo |
| Recombinant DNA reagent | bovine-Gbeta (plasmid) | *Furukawa et al., 1998* (PMID: 9651354) | | gift from Dr. Toshihide Nukada |
| Recombinant DNA reagent | bovine-Ggamma (plasmid) | *Furukawa et al., 1998* (PMID: 9651354) | | gift from Dr. Toshihide Nukada |
| Recombinant DNA reagent | pAnap (plasmid) | *Chatterjee et al., 2013* (PMID: 23924161) | Addgene: plasmid #48696 | Obtained from Dr. Peter G. Schultz through Addgene |
| Chemical compound, drug | 3-(6-acetyln aphthalen-2-ylamino)—2-aminopropanoic acid (Anap) | FutureChem | FC-8101 | |
| Chemical compound, drug | Biomol Green Reagent | ENZO Life Sciences | BML-AK111-0250 | |
| Chemical compound, drug | PtdIns-(4,5)-$P_2$ (1,2-dipalmitoyl) sodium salt (di-C16-PI(4,5)$P_2$) | Cayman | 10008115 | |
| Chemical compound, drug | PtdIns-(3,4,5)-$P_3$(1,2-dipalmitoyl) sodium salt (di-C16-PI(3,4,5)$P_3$) | Cayman | 64920 | |
| Chemical compound, drug | 1-Palmitoyl-2-Oleoyl-sn-Glycero-3-(Phospho-L-Serine) sodium salt (POPS) | Avanti Polar Lipids | 840034C | |

## cDNAs and in vitro synthesis of cRNAs

The following cDNAs were used: *Ciona intestinalis*-VSP (Ci-VSP) and bovine G-protein $\beta_1$ and $\gamma_1$ in pSD64TF; mouse-Kir3.2d (GIRK2d), rat-Kv7.2 (KCNQ2), and rat-Kv7.3 (KCNQ3) in pGEM-HE for expression in *Xenopus* oocytes; human-PTEN (hPTEN), and the cytoplasmic catalytic region (CCR) of Ci-VSP (residues 248–576) in pGEX6P1 for the in vitro phosphatase activity assay (*Murata et al., 2005*; *Iwasaki et al., 2008*; *Kurokawa et al., 2012*; *Sakata and Okamura, 2014*). A Plasmid encoding tRNA and aminoacyl-tRNA synthetase for Anap incorporation was obtained from Dr. Peter G.

Schultz through Addgene [pAnap: plasmid #48696, (*Chatterjee et al., 2013*) and it was identical to that used in our earlier study (*Sakata et al., 2016*).

All point mutations at Gln-208, Gly-214, Cys-363, Leu-284, Phe-285, and Lys-555 in Ci-VSP or Val-45 and Tyr-46 in hPTEN were generated using a Quik Change Site-Directed Mutagenesis kit (Stratagene, USA) or PrimeSTAR Mutagenesis Basal kit (Takara Bio, Japan).

cDNA plasmids for RNA synthesis were linearized with XbaI (for pSD64TF) or NotI (for pGEM-HE). cRNAs were synthesized using a mMESSAGE mMACHINE SP6 (for pSD64TF) or T7 (for pGEM-HE) transcription kit (Ambion, Inc., Austin, TX, USA). The purified cRNAs were stored at −80°C.

## In vitro phosphatase activity assay

GST-fusion proteins of full-length hPTEN and the CCR of Ci-VSP (residues 248–576) were expressed in *Escherichia coli* (JM109: Promega, USA). They were allowed to bind to a GST-column (Glutathione Sepharose 4B: GE Healthcare, USA) and washed. Then, GST-fusion proteins were digested from GST-tags using PreScission Protease (GE Healthcare, USA) at 4°C overnight. The flow-through was collected. The purified proteins were stored at −80°C. The concentrations of these proteins were determined by UV absorbance at 280 nm. Purity and aggregation were estimated by SDS-PAGE and gel-filtration chromatography (Superdex200: GE Healthcare, USA).

The malachite green assay was performed as described previously (*Iwasaki et al., 2008*; *Matsuda et al., 2011*; *Kurokawa et al., 2012*). 1,2-Dipalmitoyl-sn-glycero-3- phosphatydilinositol-3,4,5-triphosphate and −4,5-bisphosphate [PIPs: di-$C_{16}$-PI(3,4,5)$P_3$, di-$C_{16}$-PI(4,5)$P_2$: Cayman, USA] and 1-palmitoyl- 2-oleoyl- sn-glycero-3- phosphatidylserine (POPS: Avanti Polar Lipids, USA) were used. The reactions were initiated by the addition of 1 µg of recombinant hPTEN or the CCR of Ci-VSP at 23°C in 10 µL buffer solution [100 mM Tris-HCl (pH8.0)] including 1.0 nmol PIPs [di-$C_{16}$-PI(3,4,5)$_3$: hPTEN, di-$C_{16}$-(PI(4,5)$P_2$: Ci-VSP] and 9.0 nmol POPS. After 30 min of incubation, 100 mM NEM was added to terminate the reaction, and the mixture was centrifuged to remove aggregates. BIOMOL Green reagent (Enzo Life Sciences, USA) was added to the supernatants and incubated at 37°C to 30 min, and then the $OD_{620}$ was measured with a microplate reader (MultiScan FC: Thermo Fisher Scientific, USA). Specific phosphatase activities were calculated from the $OD_{620}$, which indicated the amount of phosphate released in the solution.

## Preparation of oocytes

Oocytes were prepared as described previously (*Sakata and Okamura, 2014*). Briefly, *Xenopus* oocytes were collected from frogs anaesthetized in pure water containing 0.2% ethyl 3-aminobenzoate methanesulfonate salt (Sigma-Aldrich, USA). The oocytes were defolliculated using type I collagenase (1.0 mg/mL, Sigma-Aldrich, USA) and injected with cDNA or cRNA as described below. Injected oocytes were incubated for 2–3 days at 18°C in ND96 solution. All experiments were carried out following the guidelines of the Animal Research Committees of the Graduate School of Medicine of Osaka University.

## Electrophysiology

To measure voltage dependent Ci-VSP phosphatase activities, cRNA of the WT or mutant Ci-VSP were co-injected with that of Kir3.2d, Gβ, and Gγ into oocytes (0.1–0.2: 0.05: 0.05: 0.05 ng/nL, 50 nL injection volume). The injected oocytes were incubated for 2–3 days at 18°C in ND96 solution (5 mM Hepes, 96 mM NaCl, 2 mM KCl, 1.8 mM $CaCl_2$, and 1 mM $MgCl_2$ (pH 7.5), supplemented with gentamycin and pyruvate). The macroscopic current was recorded under a two-electrode voltage clamp using an Oocyte Clamp amplifier (OC-725C: Warner Instruments, USA). Stimulation, data acquisition, and analysis were performed on a Macintosh computer using an ITC-16 AD/DA converter and Pulse software (HEKA Electronik, Germany). Intracellular glass microelectrodes were filled with 3 M KCl. The microelectrode resistances ranged from 0.5 to 1.0 MΩ, and the ND96 solution was utilized as a bath solution. Leak subtraction by the P over N protocol was not performed, except for the measurement of 'sensing' currents.

## Voltage clamp fluorometry (TMRM and Anap)

For voltage clamp fluorometry with tetramethylrodamine-maleimide (TMRM: Invitrogen, USA), cRNA of Ci-VSP Q208C or the G214C mutant was injected into oocytes (0.1–0.2 ng/nL, 50 nL), followed by

2–3 days of incubation at 18°C in ND96 solution. The Ci-VSP mutants expressed in oocytes were labeled with 10 μM TMRM for 30 min in the dark. Then, the oocytes were washed twice with ND96 solution.

Oocytes were imaged using an IX71 inverted microscope (Olympus, Japan) equipped with a 10 × 0.3 N.A. or 20 × 0.75 N.A. objective lens and a mercury arc lamp under a two-electrode voltage clamp (TEVC; Oocyte Clamp amplifier (OC-725C): Warner Instruments, USA) to control oocyte membrane potential. BP535-555, BP570-625, and DM565 (Olympus, Japan) were used for the excitation filter, emission filter, and dichroic mirror, respectively. Fluorescence was detected using one PMT (H10722-20; Hamamatsu Photonics, Japan). The output of the PMT was digitized using an AD/DA converter (Digidata1440A: Molecular Device, USA) with a TEVC setup. ND96 was utilized as a bath solution. The amount of fluorescence bleaching was not subtracted from the values determined. Averaged traces from 4 to 8 traces were shown and were digitally filtered at 300 Hz.

Voltage clamp fluorometry with 3-(6-acetylnaphthalen-2-ylamino)−2-aminopropanoic acid (Anap) was performed as described previously (*Sakata et al., 2016*). For Anap incorporation, 20 nL of pAnap DNA solution (10 ng/μL) was injected into the nucleus of oocytes. Oocytes placed in Terasaki plates were centrifuged at 1500 rpm for 10–15 min. One day later, 1 mM Anap (FutureChem, Korea) and cRNA encoding Ci-VSP mutants, in which Lys-555 was mutated to a TAG codon (0.1–0.5 ng/nl), were mixed at a 1:1 ratio, and 50 nL of the mixture was injected into the oocytes under a dim red light. The oocytes were then incubated for 2–3 days in the dark. Equipment used to measure Anap signals was the same as for TMRM measurement, except for the following differences in the mirrors and PMTs. BP330-385 and DM400 (Olympus, Japan) were used for the excitation filter and dichroic mirror, respectively, and the fluorescence signal was separated by a second dichroic mirror (DM458: Olympus, Japan) and detected using two PMTs with emission filters (BP420-460 and BP460-510: Olympus, Japan). ND96 was utilized as the bath solution, and we did not correct for fluorescence bleaching. Averaged traces from 16 traces were shown and were digitally filtered at 300 Hz.

## Data analysis

Data were analyzed using Excel (Microsoft, USA), Clampfit (Molecular Device, USA) and Igor Pro (WaveMetrics, USA) software. The rate constant of inward $K^+$ current decay, which was defined as the voltage dependent phosphatase activity, was obtained by fitting the time-dependent normalized current with a single exponential. Data were analyzed statistically, and error bars depict means ±S.D. Statistical significance was assessed with two-tailed Student's *t*-test.

## MD simulation

All MD simulations were performed using the GROMACS 4.5.4 suite (*Hess et al., 2008*). In our simulation analyses, PI(3,4,5)P$_3$ stands for 1-stearoyl-2-arachidonoyl-sn-glycero-3- phosphatidylinositol-3,4,5-trisphosphate.

## CG simulations

In the CG simulations, the Martini 2.1 force field was used (*Marrink et al., 2007*; *Monticelli et al., 2008*) in combination with the polarizable water (PW) model developed by Yesylevskyy et al. (*Yesylevskyy et al., 2010*). A developer-provided script (Martinize) was used to assign CG particles to the peptide backbone and side chains of the crystal structure of the cytoplasmic catalytic region of Ci-VSP (PDB entry 3V0D covering 246–575) determined by Liu et al. (*Liu et al., 2012*). Topological parameters, including dihedral angles, were set as recommended by the Martini 2.1 developer. Both the N- and C-terminal backbone atoms of the protein were left uncharged and were represented by a P5 particle. The topology parameters for CG PI(3,4,5)P$_3$ were prepared in a manner similar to that described by Stansfeld et al. (*Stansfeld et al., 2009*); the inositol ring was represented by three SP1 particles, and two of them were bonded to one and two Qa particles, respectively, to represent the phosphate groups. Bond lengths of PI(3,4,5)P$_3$ were adjusted based on our atomistic simulation data. For the protein, an elastic network was applied to all backbone particles using a force constant of 10 kJ mol$^{-1}$ Å$^{-2}$ and a cutoff distance of 7 Å to maintain the overall secondary and tertiary structures. PI(3,4,5)P$_3$ molecules were added to the monolayer of the bilayer that was closer to the initial position of the protein. In addition to the lipid components shown in *Tables 1* and *2* Cl$^-$ and 26 Na$^+$ ions were added to the simulation box for the POPC and POPC/PI(3,4,5)P$_3$ sets, respectively, to

neutralize the total charge. For hydration, 5774 and 7069 PW particles were used for the POPC and POPC/PI(3,4,5)P$_3$ sets, respectively. Run parameters were taken from our previous analyses (*Nishizawa and Nishizawa, 2014*). Briefly, the relative dielectric coefficient was set at 2.5. The Lennard-Jones interactions smoothly shifted to zero between 0.9 to 1.2 nm, and the electrostatic interactions shifted to zero between 0 to 1.2 nm. The integration time step was set at 20 fs, the temperature was held at 320 K using the Berendsen algorithm (*Berendsen et al., 1984*), and the pressure was controlled at 1 bar using semi-isotropic coupling with the Berendsen algorithm, with and without the compressibility of the $3 \times 10^{-4}$ bar$^{-1}$. Bond lengths were restrained using LINCS (*Hess et al., 1997*), and the protonation states of the titratable side chains of amino acids were the same as those in the atomistic (AT) simulations. The effective CG simulation time was obtained by multiplication by a factor of four (*Marrink et al., 2007*).

## Atomistic simulations

For the AT simulations, united-atom Berger parameters (*Berger et al., 1997*) were used for POPC membranes in combination with the OPLS-AA protein force field after employing the adaptation procedure designed by C. Neale (*Chakrabarti et al., 2010*). SPC (simple point charge) water (*Berendsen et al., 1984*), and the crystal structure of the cytoplasmic catalytic region of Ci-VSP (PDB entry 3V0D) was used. The N- and C-termini were left uncharged using NH$_2$ and COOH groups, respectively. The AT topology parameters for PI(3,4,5)P$_3$ were prepared by grafting of the arachidonate parameters, as described by Ollila et al. (*Berger et al., 1997*; *Ollila et al., 2007*), onto the Berger parameters of the phosphatidyl group, whereas the headgroup parameters were mainly adopted from the Berger lipid parameters; however, the partial charges were updated based on a previous paper (*Lupyan et al., 2010*). All of the 3'-, 4'-, and 5'-phosphate groups were assigned a charge of −2 and, in total, the PI(3,4,5)P$_3$ model had a charge of −7. The initial structure of PI(3,4,5) P$_3$ was prepared using the original version of Winmoster provided by N. Senda (https://winmostar. com/jp/). In addition to the lipids shown in *Table 1*, the POPC systems contained 18500 water, 2 Cl, and the CCR of Ci-VSP, whereas the POPC/PI(3,4,5)P$_3$ systems contained 21882 water, 26 Na, and the CCR. The CCR was initially placed such that the center of mass of the CCR was 2.6–3.4 nm away from the layer of POPC phosphorus atoms (*Table 1*). Run parameters, including temperature control at 323 K, were those recently described (*Nishizawa and Nishizawa, 2018*), with the following differences. The bond lengths of lipids and proteins were restrained with LINCS (*Hess et al., 1997*) and those of water with SETTLE (*Miyamoto and Kollman, 1992*). The particle-mesh Ewald algorithm was used (*Darden et al., 1993*), with a real-space cutoff of 1.4 nm and minimal grid size of 0.12 nm.

## Productive orientation

In our preliminary AT simulations, all snapshots in which both the C2 domain and PD bound to the bilayer exhibited a similar orientation (data not shown), which was consistent with the previous studies on PTEN and VSP (*Shenoy et al., 2012*; *Lumb and Sansom, 2013*; *Kalli et al., 2014*). With this orientation, the membrane-binding of the loop containing Lys-516 and Arg-520 (C2 domain β-strand 6–7 loop that corresponds to PTEN's CBR3 loop) and the loop containing Lys-555 and Lys-558 (β-strand 8–9 loop) restrained the rotational motions of the C2 domain around the axis that links the center of mass of the C2 domain to that of the PD. In this orientation, the C$_\beta$ atom of His-332 and the C$_\alpha$ atom Lys-367 in the catalytic center reside at nearly equal distances (with a difference <0.3 Å) from the bilayer (*Figure 2—figure supplement 1*). We considered this orientation to correspond to the productive orientation discussed previously (*Kalli et al., 2014*). For both present CG and AT simulations, we defined the productive orientation as the orientation with which the vector product of the vector 1 (the vector that arise from the C$_\alpha$ atom of Arg-281 of the PD to the middle point between the Lys-516 C$_\alpha$ and Arg-520 C$_\alpha$ atoms) and the vector 2 (the vector that arises from the Arg-281 C$_\alpha$ to the middle point of the Lys-555 C$_\alpha$ and Lys-558 C$_\alpha$ atoms) (*Figure 2—figure supplement 1*) forms an angle of 0–30° with the z-axis.

## Binding

To facilitate discussion of simulation results, it is important to define the 'binding' of proteins to phospholipid membranes. To help intuitive understanding, in the present simulation analyses we chose to define the 'binding' based on the distance of the protein atom with the highest

z-coordinate (which we refer to as the 'top atom') from the phosphorus layer of the inner (lower) monolayer of the bilayer. We further chose to use distinct definitions of binding between the POPC and the POPC/PI(3,4,5)P$_3$ sets to allow for the effect of the phosphate groups linked to the inositol ring to prevent protein atoms from directly contacting with phosphorylcholine groups. For the AT POPC and POPC/PI(3,4,5)P$_3$ sets, we regarded the protein as being in the bound state when the top atom was located at 0 and $-4$ Å or at a higher z-position relative to the phosphorus layer, respectively. For the CG POPC and POPC/PI(3,4,5)P$_3$ sets, the bound state was similarly defined using the cutoff of $-2$ and $-12$ Å, respectively.

## Acknowledgments

We thank Dr. Yoshihisa Kurachi (Osaka University) for providing the Kir3.2d plasmid, Drs. David McKinnon (Stony Brook University) and Koichi Nakajo for giving us Kv7.2 and Kv7.3 plasmids, Dr. Toshihide Nukada for providing G protein β- and γ-plasmids (*Furukawa et al., 1998*), and Dr. Laurinda A Jaffe for critical reading of the manuscript.

## Additional information

### Funding

| Funder | Grant reference number | Author |
|---|---|---|
| Japan Society for the Promotion of Science | JP15K18516 | Akira Kawanabe |
| Japan Society for the Promotion of Science | 16H02617 | Yasushi Okamura Kazuhisa Nishizawa |
| Japan Society for the Promotion of Science | 25253016 | Yasushi Okamura Atsushi Nakagawa Souhei Sakata |
| Core Research for Evolutional Science and Technology | JPMJCR14M3 | Atsushi Nakagawa Yasushi Okamura |
| Ministry of Education, Culture, Sports, Science and Technology | 15H05901 | Yasushi Okamura Atsushi Nakagawa |

The funders had no role in study design, data collection and interpretation, or the decision to submit the work for publication.

### Author contributions

Akira Kawanabe, Conceptualization, Formal analysis, Funding acquisition, Validation, Investigation, Visualization, Writing—original draft; Masaki Hashimoto, Tomoko Yonezawa, Yuka Jinno, Investigation; Manami Nishizawa, Formal analysis; Kazuhisa Nishizawa, Conceptualization, Formal analysis, Funding acquisition, Validation, Writing—original draft; Hirotaka Narita, Conceptualization, Data curation; Souhei Sakata, Conceptualization, Methodology, Writing—review and editing; Atsushi Nakagawa, Conceptualization, Writing—review and editing; Yasushi Okamura, Conceptualization, Supervision, Funding acquisition, Project administration, Writing—review and editing

### Author ORCIDs

Akira Kawanabe (iD) http://orcid.org/0000-0002-1094-6209
Kazuhisa Nishizawa (iD) http://orcid.org/0000-0002-7642-6054
Atsushi Nakagawa (iD) http://orcid.org/0000-0002-1700-7861
Yasushi Okamura (iD) http://orcid.org/0000-0001-5386-7968

### Ethics

Animal experimentation: All experiments were carried out following the guidelines of the Animal Research Committees of the Graduate School of Medicine of Osaka University.

Decision letter and Author response
Decision letter https://doi.org/10.7554/eLife.41653.045
Author response https://doi.org/10.7554/eLife.41653.046

## Additional files

### Supplementary files

• Transparent reporting form
DOI: https://doi.org/10.7554/eLife.41653.043

### Data availability

All data generated or analysed during this study are included in the manuscript and supporting files. Source data files have been provided for Figures 4,5,6,7,8,9 and 10.

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
