## [Decision Letter]

Thank you for submitting your article "The hydrophobic nature of a novel membrane interface regulates the enzyme activity in voltage-sensing phosphatase" for consideration by *eLife*. Your article has been reviewed by two peer reviewers, and the evaluation has been overseen by a Reviewing Editor and Richard Aldrich as the Senior Editor. The reviewers have opted to remain anonymous.

The reviewers have discussed the reviews with one another and the Reviewing Editor has drafted this decision to help you prepare a revised submission.

Summary:

The authors show data in support of the role of two hydrophobic residues in VSP for regulating the phosphates activity of VSD. MD simulations show that two hydrophobic residues allow better contact with the membrane and that nearby hydrophilic residues prevent the insertion of the hydrophobic residues in the hydrophobic core of the membrane, so that there is some flexibility of the PD. Experimental results show that the hydrophobic nature of these residues promotes the phosphatase activity and also affects the VSD motion. The data and the story are convincing and complete.

Essential revisions:

1) Subsection “Retrograde effects of mutations in the hydrophobic spine on VSD motion”, second paragraph, "The normalized F-V relationship of L284F/G214C-TMRM shows that the fluorescence reached the -1 level over 100 mV as in WT/G214C-TMRM, whereas it did not reach the -1 level at greater than 100 mV in L284Q/G214C-TMRM and F285Q/G214C-TMRM": I don't understand this sentence. You mean that the FV didn't saturate at voltage >100 mV for the mutants, but it saturated for WT?

2) Subsection “Retrograde effects of mutations in the hydrophobic spine on VSD motion”, fourth paragraph: Insert "for WT", because you are not measuring at the peak for mutants. It is not clear what you are quantifying by measuring at this time and why? Just to show it is different at this particular time point?

3) Subsection “Retrograde effects of mutations in the hydrophobic spine on VSD motion”, fourth paragraph: You are mixing phases in time and voltage. What do you really mean at > 100 mV? Along the voltage axis the first two phases have already occurred at <100mV.

4) Subsection “Retrograde effects of mutations in the hydrophobic spine on VSD motion”, fourth paragraph: Is the third component the only one connected to coupling? If so, why is the first and second altered at V >100mV also?

5) Subsection “Analysis of the fluorescence of an unnatural amino acid, Anap, suggests two-step activation of the CCR”, second paragraph: This is not a clear description of what happens. A relative larger fluorescence signal at higher voltages (meaning that the F saturates at a less positive voltage) should give a leftward shift. As written right now, it sounds like there are two effects.

6) Discussion section, first paragraph, "provide high movability": It seems like it is the hydrophilic residues that provides mobility, not the hydrophobic residues which would anchor it more to the membrane.

7) Subsection “The hydrophobic spine controls the later transition of enzyme activation”, fourth paragraph: It is not clear how you distinguish between effect directly on VSD of mutations and retrograde coupling.

8) Subsection “Mechanistic insights into coupling between the VSD and CCR”, end of first paragraph: This sentence is not clear, tell what was found earlier and describe why or why not consistent with present data.

9) Subsection “Mechanistic insights into coupling between the VSD and CCR”, second paragraph, "membrane-tethered β2 subunit suggested that Ca_V_ binds": What are β2 and Ca_V_? This is not explained.

10) Figure 11. What happens in VSD between middle and right panel? Should voltage do anything here? If so, shouldn't S4 move further out then?

11) No evidence for two different catalytic rates were observed. Multiple VSP states have been reported by other scientists, including a report from these authors (Sakata and Okamura, 2014), revealing two VSD states when testing VSD mutants but that publication does not show a difference in the catalytic activity. Experimental evidence for two catalytic states, low and high activity is needed.

(Below is the alternative suggested by the other reviewer:

I think to measure the catalytic activity with such precision as to be able to detect the low activity state is hard. This is because there will be a monotonic increase in the amount of VSP in the high activity state with increasing voltage and the voltage range for this increase and the voltage range for the proposed low activity is partly overlapping. One possibility is to have them remove the low activity of the intermediate state and just claim that the hydrophobic spine increases the catalytic activity by just altering the population in the high activity state. I don't think to explain their data, that it is necessary for their model to have the low activity in the intermediate state (it could just be no activity in the intermediate state). So my recommendation is to remove this if they don't have more data for the low activity. Removing this will not detract from the rest of their findings or conclusions.)

12) Both the Abstract and the Discussion emphasize a two state active model for VSP based on this data. While the Anap data clearly shows two states that go to one state with the Q mutations, the TMRM data is much less clear. Previous publications have all fit the G214C-TMRM data with a single Boltzmann. I would like to see residuals or other analysis that a double Boltzmann is required to accurately fit the data. I'm also concerned that two states in a fluorescence trace (whether TMRM or Anap) is interpreted as low and high activity state.

13) Regarding the interpretation of the electrophysiology and fluorescence, one of the conclusions the authors state is that there is a preference for aromaticity in the 284 position. Many of the comparisons between WT and L284F show that they are barely or only slightly different from each other. The hydrophobicity interpretation is a much stronger conclusion.

14) I have one concern regarding how the MD data is presented. Overall, description of the MD simulations was difficult to follow. I bounced back and forth between all of the supplementary figures to be able to understand how the experiments were done and how they were being interpreted. The low resolution of the supplementary figures made this more challenging as well. One aspect that confused me was the units. The MD simulations are in nm and the text in the Results section states binding was observed in Figure 2—figure supplement 2H for example. The "position from lower monolayer P atoms" for panel H range from -4 to -2 nm or -40 to -20 angstroms. The Materials and methods state that the binding is defined as -2 and -12 Å in the CG set. This range does not agree with the what is written in the Results section or the data. Is one of these units or values typos? Further explanation is needed to understand why the authors chose 0 and -4 Å for the AT simulations and another set of numbers for the CG set as well.

---

## [Author Response]

Essential revisions:1) Subsection “Retrograde effects of mutations in the hydrophobic spine on VSD motion”, second paragraph, "The normalized F-V relationship of L284F/G214C-TMRM shows that the fluorescence reached the -1 level over 100 mV as in WT/G214C-TMRM, whereas it did not reach the -1 level at greater than 100 mV in L284Q/G214C-TMRM and F285Q/G214C-TMRM": I don't understand this sentence. You mean that the FV didn't saturate at voltage >100 mV for the mutants, but it saturated for WT?

We apologize that our sentences were not clear. We changed the sentence into the following, “The normalized F-V relationship of L284F/G214C-TMRM shows that the fluorescence saturated at 150 mV as in WT/G214C-TMRM, whereas it did not saturate at 150 mV in L284Q/G214C-TMRM and F285Q/G214C-TMRM”.

2) Subsection “Retrograde effects of mutations in the hydrophobic spine on VSD motion”, fourth paragraph: Insert "for WT", because you are not measuring at the peak for mutants. It is not clear what you are quantifying by measuring at this time and why? Just to show it is different at this particular time point?

Thank you for pointing this out. We followed the style of the previous reports (Kohout et al., 2010; Grimm et al., 2016) so that we could compare our results with their results. In the Figure 8—figure supplement 5, we illustrated the time point as a black arrow. We changed the sentence into, “The averages of the normalized fluorescence changes of Q208C-TMRM at the time point of peak signal of the 200-mV trace from WT (black arrow heads in Figure 8—figure supplement 5A) were plotted against the membrane potential”.

3) Subsection “Retrograde effects of mutations in the hydrophobic spine on VSD motion”, fourth paragraph: You are mixing phases in time and voltage. What do you really mean at > 100 mV? Along the voltage axis the first two phases have already occurred at <100mV.

We apologize for this confusion. The sentence was not correct due to our mistake. We corrected the sentence into, “There are no significant differences among WT and mutants in the first and second phases of the F-V relationship observed at voltages <100 mV.”

4) Subsection “Retrograde effects of mutations in the hydrophobic spine on VSD motion”, fourth paragraph: Is the third component the only one connected to coupling? If so, why is the first and second altered at V >100mV also?

We apologize for our error and corrected it as described above to the comment 3.

5) Subsection “Analysis of the fluorescence of an unnatural amino acid, Anap, suggests two-step activation of the CCR”, second paragraph: This is not clear description of what happens. A relative larger fluorescence signal at higher voltages (meaning that the F saturates at a less positive voltage) should give a leftward shift. As written right now, it sounds like there are two effects.

Thank you for your valuable comment. We consider that fluorescence changes do include both of two effects, since the statistic analysis in Figure 10 and Table 2 showed significant differences in the amplitude of fluorescence intensity and voltage dependence. We now added the following sentence, “Analysis of F-V relationships revealed that both the amplitude increase and leftward shift of voltage dependence underlie the larger fluorescence increase of the second component (Figure 10B and Table 2).”

6) Discussion section, first paragraph, "provide high movability": It seems like it is the hydrophilic residues that provides mobility, not the hydrophobic residues, which would anchor it more to the membrane.

Thank you for this suggestion. We agree that a careful discussion benefits here. As the reviewer points out, and based on the MacCullum et al. (2008), we think that in the layer of the lipid head groups near the phosphorus atoms, not only the hydrophobic but also the hydrophilic residues help movability of the hydrophobic spine. So, we have now changed this sentence of the Discussion as follows: “Our MD simulations of the CCR suggested that the combination of the hydrophobic spine consisting of two conserved hydrophobic amino acids (Leu-284 and Phe-285), and the flanking hydrophilic residues provides high mobility to the PD in and near the membrane/water interface.”

7) Subsection “The hydrophobic spine controls the later transition of enzyme activation”, fourth paragraph: It is not clear how you distinguish between effect directly on VSD of mutations and retrograde coupling?

Thank you for your insightful comment. We guess that actual molecular mechanisms by hydrophobic spine will still remain unclear until the protein structure of the full-length VSP is solved. We think that the hydrophobic spine is critical for anterograde coupling from the voltage sensor domain to enzyme activity as proposed in this paper. Effect of mutation of the hydrophobic spine on the voltage sensor motion is significant as we demonstrated in VCF results of G214C-TMRM. This retrograde coupling serves as a good quantitative indicator of coupling between the voltage sensor and enzyme region based on the tight coupling between the two regions. We think that effect on the motion of voltage sensor by mutation in the hydrophobic spine is due to the change of coupling condition between the two regions. It is unlikely that our results of changes of voltage-dependent enzyme activity upon mutation of the hydrophobic spine are due to the altered motion of the voltage sensor, since there is no evidence that hydrophobic spine stays next to the S4 or interact with other helices of the voltage sensor.

8) Subsection “Mechanistic insights into coupling between the VSD and CCR”, end of first paragraph: This sentence is not clear, tell what was found earlier and describe why or why not consistent with present data.

Thank you very much for your helpful comment. This part was revised so that the meaning is clearer. We previously reported that the distance between the CCR and membrane was not significantly changed. However, we only estimated the distance from the C2 domain in our previous report. In the present study, we assume that the distance between the phosphatase domain and membrane is changed, whereas the C2 domain stays anchored to the membrane irrespective of the state of VSD. Then, we changed the sentence, “This result may seem to conflict with our previous results that the distance between the CCR and plasma membrane was not significantly changed upon voltage-induced activation based on the measurement by FRET between Anap and a membrane-embedded FRET quencher, dipicrylamine (DPA)”.

9) Subsection “Mechanistic insights into coupling between the VSD and CCR”, second paragraph, "membrane-tethered β2 subunit suggested that Ca_V_ binds": What are β2 and Ca_V_? This is not explained.

In this sentence, we mentioned voltage-gated Ca^2+^ channels, “Ca_V_”. We modified the sentence, “Studies of the membrane-tethered β2 subunit of voltage-gated Ca^2+^ channel (Ca_V_) suggested that Ca_V_ binds to the plasma membrane…”

10) Figure 11. What happens in VSD between middle and right panel? Should voltage do anything here? If so, shouldn't S4 move further out then?

We think that VSD has two-step activation because the F-V curve of G214C-TMRM could be fitted by the sum of two Boltzmann distributions. We therefore redrew the cartoon of the VSP in the middle panel of Figure 11, making it clearer that the position of S4 is in the intermediate state.

11) No evidence for two different catalytic rates were observed. Multiple VSP states have been reported by other scientists, including a report from these authors (Sakata and Okamura, 2014), revealing two VSD states when testing VSD mutants but that publication does not show a difference in the catalytic activity. Experimental evidence for two catalytic states, low and high activity is needed.(Below is the alternative suggested by the other reviewer:I think to measure the catalytic activity with such precision as to be able to detect the low activity state is hard. This is because there will be a monotonic increase in the amount of VSP in the high activity state with increasing voltage and the voltage range for this increase and the voltage range for the proposed low activity is partly overlapping. One possibility is to have them remove the low activity of the intermediate state and just claim that the hydrophobic spine increases the catalytic activity by just altering the population in the high activity state. I don't think to explain their data, that it is necessary for their model to have the low activity in the intermediate state (it could just be no activity in the intermediate state). So my recommendation is to remove this if they don't have more data for the low activity. Removing this will not detract from the rest of their findings or conclusions.)

Our idea of low enzyme activity for the middle state in our scheme mainly came from our findings of Anap fluorescence pattern from low-enzyme activity mutants (L284Q). The main conclusion in our paper is that there are at least two steps of enzyme activation until the protein reaches a full enzyme activation, and we think how enzyme is active in the middle state still needs further careful analysis. As pointed out, we did not provide conclusive experimental evidence for two reaction rates of enzyme activity of Ci-VSP although we suggested that in our scheme of enzyme activation in the previous version.

In this version, we studied Anap fluorescence from more constructs of mutants with varied enzyme activity, L284K and L284T. L284K shows lower enzyme activity than L284Q. L284T shows moderate enzyme activity between L284Q and WT. The low-enzyme activity mutant L284K showed the F-V curve with single Boltzmann function with negativity as L284Q. On the other hand, L284T showed mainly 1^st^ component but a notch was noted at higher voltage, suggesting the presence of a small but significant later component. This is consistent with our idea that the second component is critical for robust enzyme activity. These new results are now included as Figure 10—figure supplement 2.

Nevertheless, we need to admit that we cannot completely exclude a possibility that the enzyme at the middle state (partially activated state) has no enzyme activity, since we have not provided evidence for low enzyme activity of the wild type, due to overwhelming robust enzyme activity of the second component (as discussed by another reviewer). This test must require simultaneous detection of enzyme activity and of protein conformation at the single protein level in the membrane which is not available so far. Therefore, some phrases of “low enzyme activity in the middle state” were deleted from the text, and the reason for our idea of “low enzyme activity in the middle state” was more clearly stated in the figure legend and modified the phrase in the figure from “Inactive -> Low activity -> High activity” to “Resting state -> Partially-activated state -> Fully-activated state” (Figure 11). We also weakened the phrase “showed” to “suggest” in the Abstract.

12) Both the Abstract and the Discussion emphasize a two state active model for VSP based on this data. While the Anap data clearly shows two states that go to one state with the Q mutations, the TMRM data is much less clear. Previous publications have all fit the G214C-TMRM data with a single Boltzmann. I would like to see residuals or other analysis that a double Boltzmann is required to accurately fit the data. I'm also concerned that two states in a fluorescence trace (whether TMRM or Anap) is interpreted as low and high activity state.

Thank you for your important comment. As the reviewers noted, in all of previous reports on VCF data of G214CTMRM, the F-V relationship could be fitted by a single Boltzmann. To confirm the reproducibility of our data, therefore, we performed more experiments of examining the fluorescence changes of G214C-TMRM. As in the results which we previously showed, a combination of two Boltzmann functions fit the data better again than single component of Boltzmann function in the new data set as shown as the modified figure showing analyses of three individual cells (Cell 1 and 2 are from the previous data and Cell 3 is from the new data: Figure 10—figure supplement 1). Kinetics of fluorescence changes upon activation could be fitted by double exponential function as observed in previous reports, suggesting that VSP takes two-step transition. However, as pointed out, contribution of the lower-voltage component (red curve in Figure 10—figure supplement 1) in fluorescence change of G214C-TMRM to the whole F-V curve is much less clear than in Anap study. This was probably the reason why in previous publications single Boltzmann function appeared to be sufficient to fit VCF data.

Regarding the comment “I'm also concerned that two states in a fluorescence trace (whether TMRM or Anap) is interpreted as low and high activity state”please see our above response to the comment 11.

13) Regarding the interpretation of the electrophysiology and fluorescence, one of the conclusions the authors state is that there is a preference for aromaticity in the 284 position. Many of the comparisons between WT and L284F show that they are barely or only slightly different from each other. The hydrophobicity interpretation is a much stronger conclusion.

We believe that aromaticity in the hydrophobic spine has significant contribution to its role in regulation of enzyme activity for the following reasons:

1) Voltage-dependent enzyme activity of mutant harboring additional aromatic residue on the hydrophobic spine (on the 284^th^ residue) as estimated by GIRK channel activities was deviated to a positive direction from the linear relationship between hydrophobicity and enzyme activity as shown in the plot of Figure 5B.

2) Motion of VSD is clearly affected by addition of aromatic residue on 284^th^ as examined by voltage clamp fluorometry of TMR attached on the voltage sensor (Figure 8—figure supplement 3) through the tight coupling between enzyme and voltage sensor in VSP. We have recently obtained similar enhancement of voltage-dependent enzyme activity of the zebrafish VSP by addition of aromatic residue.

3) Using another PIP_2_ reporter, KCNQ2/3 channel, we found similar enhancement of voltage-dependent enzyme activity of L284F as compared with WT (Figure 5—figure supplement 3).

4) Elimination of hydrophobic residue on the spine remarkably reduces both the in vitro enzyme activity by the malachite green assay and in vivo enzyme activity in oocyte, whereas elimination of aromatic ring from the spine only affected the latter activity, indicating that aromatic ring has distinct effect than the hydrophobicity, as pointed out in the previous version of our manuscript.

Nevertheless, in the previous version, we have not demonstrated statistical differences of F-V data between WT and L284F. Therefore, we performed similar experiment of VCF measurement of Anap on the L284F mutant and obtained more data (summarized in Figure 9 and Table 2). Significant differences were found between the WT and L284F mutant after new data were added. These results indicate that Anap F-V curve from L284F shows a larger second component with leftward shift of voltage range than the WT. We thus conclude that aromaticity also plays an important role in Ci-VSP activity and its state transition as well as hydrophobicity.

14) I have one concern regarding how the MD data is presented. Overall, description of the MD simulations was difficult to follow. I bounced back and forth between all of the supplementary figures to be able to understand how the experiments were done and how they were being interpreted. The low resolution of the supplementary figures made this more challenging as well. One aspect that confused me was the units. The MD simulations are in nm and the text in the Results section states binding was observed in Figure 2—figure supplement 2H for example. The "position from lower monolayer P atoms" for panel H range from -4 to -2 nm or -40 to -20 angstroms. The Materials and methods state that the binding is defined as -2 and -12 Å in the CG set. This range does not agree with the what is written in the Results section or the data. Is one of these units or values typos? Further explanation is needed to understand why the authors chose 0 and -4 Å for the AT simulations and another set of numbers for the CG set as well.

We apologize for the poor presentation in our original version that caused difficulty reviewing. Now we have moved the table (that summarizes of the simulations) from the supplementary file to the main text (Table 1). This would help, as we now have only two files in total.

In terms of the low resolution of the figures, we feel that some resolution is lost upon the conversion to the PDF. Just in case, we have now improved the panels of Figure 2—figure supplement 2, 4 and 5.

Regarding the issue of Å and nm in Materials and methods and Figure 2—figure supplement 2: we actually made no mistake, but we understand that our writing was confusing. In Materials and methods and in the AT data (Figure 2—figure supplement 4), we described the 'binding', so the z-position of the 'top atom' relative to the mean z-position of the P atoms was plotted. In the Table 1 as well, to summarize the time period of the bound state, we also computed the top atom position and distance from the P-layer for all AT and CG simulations. However, in the CG trajectory data, 'the peptide com (the center of mass) z-position' was plotted (Figure 2—figure supplement 2). We favor this presentation scheme because it is customary to show the z-position of the ‘com’ of the protein in CG analyses, and, unlike AT simulations, in CG simulations the protein orientation changes widely (due to the lack of the anchoring to the VSD), and therefore the top atom often changes artifactually. For these reasons, we think to show the ‘com’ movement is a reasonable presentation scheme. In this regard, we have now added some lines in the legend for Figure 2—figure supplement 2 legend to emphasize that, in the Table 1 the top atom was used whereas in the Figure 2—figure supplement 2 ‘com’ was used.

In terms of the definition of the 'binding', we used definitions differing between the AT and CG analyses, as the reviewer points out. For example, the binding to POPC membrane was defined using 0 Å (or above) for AT and -2 Å (or above) for CG, relative to the phosphorus layer. This is because the CG model employs large particles; the Lennard-Jones potential sigma parameters (i.e., the distance at which the attractive and the repulsive forces cancel each other) for most CG beads are about 4.7 Å, and that for C atom of the AT model is 3.5 Å. In the case of the PI(3,4,5)P_3_-containig POPC membrane, we defined the 'binding' as the top atom positioning at -4 Å (or above) for AT and -12 Å (or above) for CG simulations. This large difference was based on our observation that in CG simulations, both PI(3,4,5)P_3_ molecules and protein have smooth and round surfaces, hampering PI(3,4,5)P_3_ to get into the groove of the protein (VSP), whereas in the AT simulations PI(3,4,5)P_3_ can enter into the groove of the protein enabling the tight contact between the VSP and the membrane. So, we have now just added a brief comment on these matters in the Materials and methods section.